# Rapid Prediction of Seismic Incident Angle's Influence on the Damage Level of RC Buildings Using Artificial Neural Networks

Konstantinos Morfidis [1,*] and Konstantinos Kostinakis [2]

1    Earthquake Planning and Protection Organization (EPPO-ITSAK), Terma Dasylliou,
     55535 Thessaloniki, Greece
2    Department of Civil Engineering, Aristotle University of Thessaloniki, Aristotle University Campus,
     54124 Thessaloniki, Greece; kkostina@civil.auth.gr
*    Correspondence: konmorf@gmail.com

**Abstract:** The angle of seismic excitation is a significant factor in the seismic response of RC buildings. The procedure required for the calculation of the angle for which the potential seismic damage is maximized (critical angle) contains multiple nonlinear time history analyses, each using different angles of incidence. Moreover, the seismic codes recommend the application of more than one accelerogram for the evaluation of seismic response; thus, the whole procedure becomes time consuming. Herein, a method to reduce the time required for the estimation of the critical angle based on multilayered feedforward perceptron neural networks is proposed. The basic idea is the detection of cases in which the critical angle increases the class of seismic damage compared to the class that arises from the application of the seismic motion along the buildings' structural axes. To this end, the problem is expressed and solved as a pattern recognition problem. The ratios of seismic parameters' values along the two horizontal seismic records' components, as well as appropriately chosen structural parameters, were used as the inputs of the networks. The results of analyses show that the neural networks can reliably detect the cases in which the calculation of the critical angle is essential.

**Keywords:** artificial neural networks; pattern recognition; reinforced concrete buildings; seismic damage; rapid assessment; seismic incident angle





## 1. Introduction

The angle of seismic excitation has been recognized for many years by the civil engineering research community as a significant factor in the seismic response of RC buildings. The level of influence of this angle on the seismic response has been proven in several published studies. Among the research papers that deal with this subject, the following are some of the most significant: MacRae and Mattheis [1] examined the impact of the angle of incidence on the inelastic response of a three-story steel-frame building due to near-fault ground motion. Athanatopoulou [2] introduced analytical formulae for the determination of the critical incident angle and the corresponding maximum elastic structural response of buildings subjected to three correlated seismic components; the application of these formulae to multistory structures has proven that the maximum value of a response quantity can be up to 180% larger than the response produced when the seismic accelerograms act along the structural axes. In another study, Rigato and Medina [3] investigated the inelastic response of asymmetric and symmetric buildings with regard to the angle of incidence by using 39 pairs of ground motion records; the results showed that the critical angle for a given response quantity depends on the fundamental period, model type, and level of inelastic behavior, and it is difficult to determine a priori, as in the case of elastic structures. In a series of studies conducted by Kostinakis et al. [4–7] and Fontara et al. [8], the influence of the seismic motion's orientation on the elastic and

inelastic seismic behavior of RC structures was investigated; they found that the structural response is strongly dependent on the strong motion's direction. Pavel and Nica [9] investigated the seismic behavior of doubly symmetric RC wall structures subjected to a series of bidirectional ground motion recordings; they found that the structural response is dependent on the orientation of the strong ground motion horizontal components. Cavdar and Ozdemir [10] examined the change in response of a seismically isolated structure as a function of ground motion orientation, and showed that using the maximum direction of ground motion does not lead to the maximum response compared to that obtained by considering different orientations of seismic records. The issue of the seismic motion's orientation from a probabilistic point of view has also been investigated by a number of researchers (see, e.g., Lagaros [11]; Giannopoulos and Vamvatsikos [12]; Vargas et al. [13]; Skoulidou et al. [14]; Skoulidou and Romão [15]), who tried to evaluate the impact of the angle of seismic incidence on the buildings' risk.

The angle of seismic excitation is defined as the angle that is formed by the orientation of the seismic input and the structural axes of buildings. In common practice, the considered orientation of the seismic input coincides with the structural axes; however, as mentioned in the above literature review, it has been proven that the most unfavorable seismic response generally arises for a different angle, which is referred to as the critical angle of the seismic excitation. The procedure for the estimation of the critical angle requires the performance of multiple analyses. In each of these analyses, the angle of seismic input is increased from 0 (which corresponds to the orientation of the structural axes of the buildings) to 360 degrees through an increment that is usually set to 5 degrees. If the evaluation of the seismic response concerns the prediction of the potential seismic damage, nonlinear time history analysis (NTHA) is required. Thus, for the estimation of the critical angle a series of NTHA must be performed. Moreover, the modern seismic codes (see, e.g., [16]) recommend the evaluation of the seismic response of buildings using more than one earthquake record. Naturally, each of the selected seismic motions can lead to a different critical angle, following the application of the above-mentioned procedure. Therefore, the whole procedure requires the implementation of a number of sets of NTHA. Beyond the stage of analysis, each of these sets includes post-processing of the extracted results for the estimation of the selected seismic damage index. Thus, the ability to avoid the procedure for the estimation of the critical angle for a seismic record is desirable. In order to achieve this, an available computational tool is the use of artificial neural networks (ANNs).

The implementation of ANNs for the solution of civil engineering problems has progressively increased over the past three decades. This fact is reflected in several published review papers [17–21]. Concentrating on the seismic assessment of RC buildings, among the numerous papers that deal with the application of ANNs, several are noteworthy. Firstly, it is necessary to mention the studies of Stephens and VanLuchene [22] and Molas and Yamazaki [23] as the first systematic attempts to use properly trained ANNs in order to rapidly predict the level of seismic damage to structures. De Stefano et al. [24] successfully studied the ability of probabilistic neural networks in the prediction of the mechanisms of seismic damage to monumental buildings. Sanchez-Silva and Garcia [25] examined the efficiency of the combined performance of multilayered feedforward neural networks and fuzzy logic theory for the earthquake damage assessment of structures using data extracted from in situ surveys after a strong earthquake. Feedforward neural networks were implemented efficiently by Lagaros and Fragiadakis [26] for the fragility assessment of steel frames. Gonzalez and Zapico [27] successfully tested the ability of multilayered perceptron networks for the reliable identification of seismic damage to multistory welded steel moment frame buildings. Lautour and Omenzetter [28] proved that the multilayered perceptron networks have the ability to reliably predict seismic damage to 2D RC buildings. Arslan [29] investigated the effects of several structural parameters on the seismic performance of regular RC buildings using multilayered perceptron networks trained with data that were created artificially, and proved the effectiveness of ANNs in the reliable prediction of the structures' seismic response. Vafaei et al. [30] successfully the studied performance

of multilayered perceptron networks in the real-time identification of earthquake-induced damage to RC shear walls. Morfidis and Kostinakis [31–33] examined the performance of the multilayered perceptron networks and the radial basis function networks in the rapid estimation of the seismic damage level using regression as well as a pattern recognition approach, and extracted very promising results.

In the present paper, the effective implementation of multilayered perceptron (MLP) networks for the rapid prediction of the influence of the seismic incident angle on RC buildings' seismic damage level (SDL) is examined. The basic idea is the application of MLP networks for the prediction of the cases in which the critical angle has a strong (or not strong) influence on the potential SDL of RC buildings. Reliably predicting the level of influence of the critical angle on the SDL for an RC building at the stage of design, the above-mentioned time-consuming procedure for the estimation of this influence can be avoided. The same is valid in case of the seismic vulnerability assessment of existing RC buildings. The problem was formulated and solved as a pattern recognition (PR) problem [34]; thus, the seismic damage was expressed by means of predefined seismic damage classes (SDC), and the MLP networks were used to predict whether the critical angle alters a building's SDC in comparison with the SDC in which the building is classified when the angle of seismic excitation is considered as equal to zero, i.e., the case in which the angle of seismic excitation coincides with the structural axes of the building. Additionally, the MLP networks were used to predict the specific SDC in which an RC building is classified when the critical angle is considered. In the above-mentioned approaches, three seismic damage classes were predefined—namely, slight, medium, and heavy—on the basis of the selected seismic damage index (SDI) (see Table 1 and Figure A4 in Appendix A). The selected SDI was the maximum interstory drift ratio (MIDR), which is used in several research papers (see, e.g., [35,36]). For the training of the MLP networks, three datasets were used: one dataset based on analyses of RC buildings without masonry infills (Bare Buildings; BB), one dataset based on analyses of RC buildings with masonry infills (Regularly Infilled Buildings; RIB), and one dataset based on analyses of RC buildings with masonry infills in all stories except for the ground story (Irregularly Infilled Buildings; IIB). The full descriptions of these three datasets, as well as the corresponding procedures for their generation, are presented in [32,33]. An extended investigation for the estimation of the optimal configuration of the MLP networks—i.e., the configuration that leads to the most reliable prediction of the influence of the incident angle—was performed for all of the above-mentioned approaches to the problem. Moreover, it must be noted that the procedure proposed in the present study gives engineers the ability to optimize the configuration of RC buildings at the stage of design by testing several alternative configurations (without the requirement for the implementation of numerus time-consuming NTHAs), and to select the one that leads to seismic response being slightly influenced by the angle of seismic excitation. The results of the present research led to the basic conclusion that the MLP networks can reliably predict the level of influence of the critical angle of seismic excitation on the seismic damage to RC buildings, in real time.

**Table 1.** Definition of 3 or 5 SDCs according to MIDR seismic damage index.

| MIDR (%) | <0.25 | 0.25–0.5 | 0.5–1.0 | 1.0–1.5 | >1.5 |
|---|---|---|---|---|---|
| SDC (5 classes) | Null | Slight | Moderate | Heavy | Destruction |
| SDC (3 classes) | Slight ("S") | | Moderate ("M") | Heavy ("H") | |
| Description | No damage or repairable damage in structural system | | Significant but repairable damage in structural system | Non-repairable damage in structural system | |

## 2. Short Theoretical Background

A short theoretical background of the scientific fields involved in the present paper is presented in this section. Firstly, some of the fundamentals that describe the problem of estimation of the critical incident angle for RC buildings are given in Section 2.1. Afterwards, some of the basic principles that concern the implementation of MLP networks for the solution of PR problems are presented in Section 2.2.

### 2.1. The Influence of Incident Angle on the Seismic Response of RC Buildings

In the vast majority of earthquake records' databases, the horizontal components of the seismic motion are given along the orientation for which they were recorded. Thus, the orientation of the recorded seismic components is predetermined by the orientation of the recording instrument (accelerograph), which is generally arbitrary with respect to the structural axes of the buildings (Figure 1) (see Beyer and Bommer [37]). In Figure 1, $\alpha_x(t)$ and $\alpha_y(t)$ represent the recorded earthquake acceleration time histories at the place of the accelerograph along its x and y axes, respectively. These accelerograms can be applied along any angle of incidence θ between the seismic components and the building's structural axes X and Y ($a_x(\theta)$ and $a_y(\theta)$).

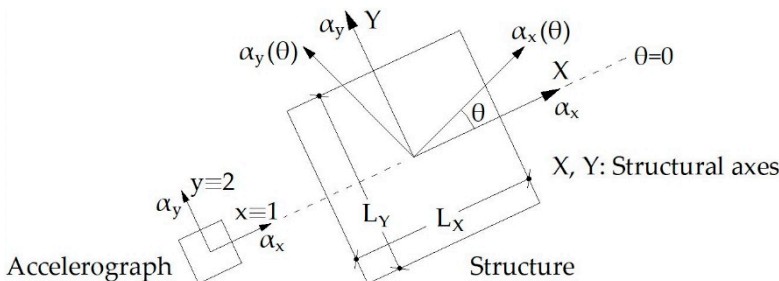

**Figure 1.** Recorded ground motion and orientation of a building's structural axes.

The seismic response of any structure is dependent on the orientation of the strong motion with regard to the structural axes. It is of great importance to note that the rotation of the axes along which the horizontal accelerograms are applied can lead to completely different structural behavior.

Until now, the orientation of the earthquake record has not caught the attention of the most modern seismic codes. Specifically, EN1998-2 [38] does not make any reference to this issue and, according to FEMA 356 [39], ASCE 41-06 [40], and ASCE 41-13 [41], the axes of the ground motion "*shall, in general, be aligned with the principal axes of the structure*". EN1998-1 [16] states that the seismic action shall "*be applied along all relevant horizontal directions*". Nevertheless, no specifications are made concerning the relevant horizontal directions—with the exception of buildings with resisting elements in two perpendicular directions, in which these two directions shall be considered as the relevant ones. NZS (Code and Supplement of the New Zealand Standard) [42] states that the seismic accelerograms are applied along the direction that will lead to the most unfavorable value of the considered response parameter—without specifying, however, how the most critical direction should be determined.

The lack of specific code provisions with regard to the axes of the strong motion input leads to the application of the horizontal earthquake components along the structural axes of the buildings. This common engineering practice can result in significant underestimation of seismic demands (see, e.g., Athanatopoulou [2]; Rigato and Medina [3]; Kostinakis et al. [4–7]; Fontara et al. [8]; Pavel and Nica [9]; Cavdar and Ozdemir [10]; Skoulidou et al. [14]; Lucchini et al. [43]; Nguyen and Kim [44]; and Roy et al. [45]). Finally, special mention must be made of the significant research of Smeby and Der Kiureghian [46] and Menun and Der Kiureghian [47], who developed the CQC3 modal combination rule, which has been successfully used for the investigation of the influence of the seismic excitations'

angle on structures' responses within the context of the linear response spectrum analysis (see, e.g., Lopez et al. [48,49]; Sessa et al. [50,51]).

### 2.2. The Solution of the PR Problem Using MLP Networks

The pattern recognition (PR) problem is one of the subjects of machine learning (ML) algorithms (see, e.g., [34,52]). The inherent ability of MLP networks to solve problems that are characterized by high nonlinearity makes them capable of effectively solving PR problems. By definition, PR is the procedure of the search for specific patterns in datasets. Between the three different general types of PR problems (i.e., supervised learning problems, unsupervised learning problems, and reinforcement learning problems), the present paper deals with a supervised learning problem. In the framework of a supervised learning PR problem, the ML algorithm attempts to correctly classify the samples of a dataset to predefined classes, which are defined by properly selected ranges of values of preselected parameters; this problem is called the classification problem. In present paper, the predefined classes are the SDCs, and the parameter that is used for their definition is the SDI (a detailed presentation of these definitions is given in Section 3).

As regards the solution of the PR problem in its classification form using MLP networks, a proper configuration of the latter is required. In Figure 2, the general form of an MLP network configured for the solution of a classification problem is presented.

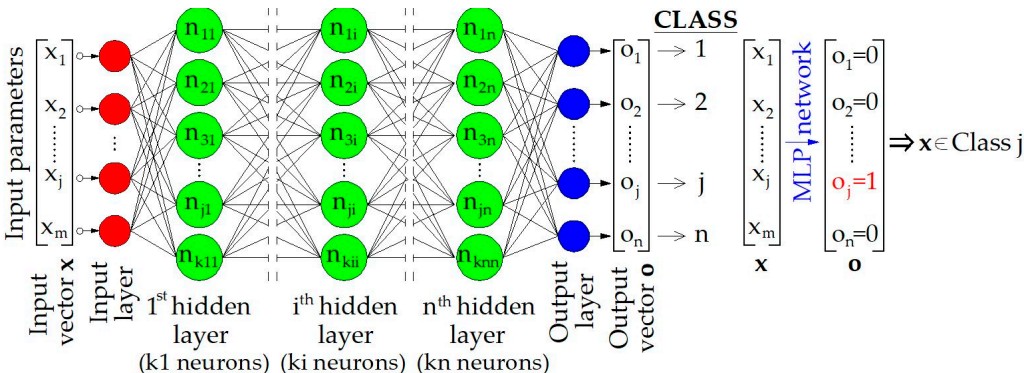

**Figure 2.** General form of an MLP network configured for the solution of classification problems.

As can be seen from Figure 2, one basic characteristic of the MLP network that is configured for the solution of a classification problem is the number of elements of the output vector **o**. This number is equal to the number of the predefined classes in which the samples **x** can be classified. Thus, for a classification problem with **n** classes, the output vector must have **n**x1 dimensions. Another basic characteristic of the modeling of the classification problem by means of MPL networks is the mapping of classes to the values of the elements of the output vector **o**. More specifically, when the network extracts an output vector **o** with $o_j = 1$ and all other elements equal to 0, the corresponding sample is classified to class j. As regards the other parameters that are required for the configuration of the MLP networks (i.e., the number of the hidden layers, the number of neurons in each hidden layer, and the activation functions), as well as the training algorithms, details are given in Section 3.

### 3. Description and Formulation of the Problem in Terms Compatible with MLP Networks

#### 3.1. General Description of the Problem and the Benefits of the Solution Using MLP Networks

The motivation for the present research is based on the idea of deploying the inherent attributes of MLP networks to extract results for multiparameter problems in real time. Thus, in the present paper, the ability of MLP networks to reliably predict the level of influence of the seismic incident angle on the SDL of new (at the stage of design) or existing RC buildings (in the framework of the seismic vulnerability assessment procedure) is

investigated. In other words, the main target of the present investigation is the proposal of a procedure based on MLP networks, which aims to detect in real time the cases for which the critical angle of the seismic excitation significantly alters the expected SDL of RC buildings compared to the SDL, which arises when the buildings are analyzed considering that the angle of seismic excitation is equal to zero (i.e., when the two horizontal components of the seismic excitation are considered to be parallel to the structural axes of the buildings; see Figure 1). The expected benefit of the proposed procedure is the avoidance of the numerous time-consuming NTHAs that are required for the estimation of the critical seismic angle, in those cases where this angle does not in fact lead to a significant increase in the SDL. Thus, it becomes feasible to rapidly identify the RC buildings whose SDL is not sensitive to the angle of seismic excitation. This rapid detection can accelerate the procedure followed for the configuration of new RC buildings at the stage of design, or the procedure for the configuration of existing RC buildings' retrofitting schemes in the framework of seismic vulnerability assessment.

*3.2. Formulation of the Problem in Terms Compatible with MLP Networks*

In order to quantify the criterion that defines the significance (or the insignificance) of the critical seismic angle's influence on the SDL, first of all, a seismic damage index (SDI) [53] must be selected. In the present paper, the MIDR index is used as the SDI (the procedure for the calculation of the MIDR is presented in Appendix A). Then, a magnification factor that reflects the increase in the SDI value when the seismic excitation angle takes its critical value $\theta_{cr}$ relative to the SDI value that corresponds to the case in which the seismic excitation angle $\theta$ is equal to zero must be defined. The general form of this factor (seismic damage index magnification factor; SDIMF) is given by means of Equation (1). A significant clarification that regards the definition of the angle $\theta = 0$ in Equation (1) must be made herein. The structural axes (which coincide with $\theta = 0$ in the present study) are not generally selected using strict rules, except in cases in which the local axes of all vertical structural members are parallel with two specific axes. However, the different definitions of structural axes in buildings do not affect the results of the proposed procedure, since the definition of the angle $\theta = 0$ is simply the reference point for the definition of the $\theta_{cr}$. Thus, if for example the initial choice of $\theta = 0$ leads to the most unfavorable seismic response (i.e., $\theta_{cr} = 0$), the proposed method will extract the conclusion that there is no need for the performance of a procedure for the calculation of the value of $\theta_{cr}$. Moreover, the proposed method leads to the answer to the question of whether the initial selection of the structural axes ($\theta = 0$) leads (or not) to the most unfavorable seismic response.

$$\text{SDIMF} = \frac{\text{Seismic Damage Index for } \theta = \theta_{cr}}{\text{Seismic Damage Index for } \theta = 0} = \frac{\text{SDI}(\theta = \theta_{cr})}{\text{SDI}(\theta = 0)} > 1.0 \qquad (1)$$

After the above definition, a characteristic value of SDIMF (value $\text{SDIMF}_c$) for the separation of the range of its values in two subdomains—namely, the subdomain of values that correspond to the significant influence of the critical incident angle on the SDI, and the subdomain of values that correspond to the insignificant influence of the critical incident angle on the SDI—must be selected; however, the verification of this selection is very difficult. Nevertheless, due to the fact that the SDIMF and SDI are related through Equation (1), it is equivalent to use a corresponding characteristic value for the latter. Thus, the problem is converted to a comparison of the SDI ($\theta = \theta_{cr}$) value with the characteristic value of SDI (value $\text{SDI}_c$), which corresponds to the $\text{SDIMF}_c$. The selection of a value for the $\text{SDI}_c$ can be verified more rationally on the basis of experimental or numerical results, which are available in the literature. More specifically, as regards the MIDR index that is used as an SDI in the present research, the limit values that define different SDCs for RC buildings are presented in [54]; these values are summarized in Table 1.

Using the values that are presented in Table 1, the definition of characteristic values for the MIDR is feasible. For example, if three SDCs are considered, as in the present paper,

then two characteristic values for the MIDR can be defined, namely, the value 0.5%, which corresponds to the limit between the slight ("S") SDC and the moderate ("M") SDC; and the value 1.0%, which corresponds to the limit between the "M" SDC and the heavy ("H") SDC. Thus, two different approaches for the formulation of the problem as a PR problem were considered in the present study:

- First Approach (Approach 1 or A1). Definition of two classes: the class of buildings for which the SDC for $\theta = 0$—i.e., SDC ($\theta = 0$)—is changed for $\theta = \theta_{cr}$ (i.e., if MIDR < 0.5% for $\theta = 0$ becomes MIDR > 0.5% for $\theta = \theta_{cr}$, or if 0.5% < MIDR < 1.0% for $\theta = 0$ becomes MIDR > 1.0% for $\theta = \theta_{cr}$), and the class of buildings for which the SDC for $\theta = 0$, SDC ($\theta = 0$), is not changed for $\theta = \theta_{cr}$ (i.e., if MIDR < 0.5% for $\theta = 0$ remains MIDR < 0.5% for $\theta = \theta_{cr}$, or if 0.5% < MIDR < 1.0% for $\theta = 0$ remains 0.5% < MIDR < 1.0% for $\theta = \theta_{cr}$). It must be noted that the case in which MIDR ($\theta = 0$) > 1.0% is not considered herein, because if a building suffers heavy damage for $\theta = 0$, its SDC cannot be changed for $\theta = \theta_{cr}$. In other words, A1 corresponds to a two-class PR problem, the solution of which leads to the answer to the question of whether the SDC of an RC building for $\theta = 0$ (SDC ($\theta = 0$)) is increased for $\theta = \theta_{cr}$ (i.e., SDC ($\theta = \theta_{cr}$) > SDC ($\theta = 0$) → significant influence of $\theta_{cr}$) or not (i.e., SDC ($\theta = \theta_{cr}$) = SDC ($\theta = 0$) → insignificant influence of $\theta_{cr}$), regardless of the SDC for $\theta = 0$ (MIDR ($\theta = 0$) < 0.5% or 0.5% < MIDR ($\theta = 0$) < 1.0%). This approach does not give additional information about the magnitude of change of SDC for $\theta = \theta_{cr}$, but simply gives the information about the change (or not) in SDC;

- Second Approach (Approach 2 or A2). In the framework of the second approach, more details about the influence of $\theta_{cr}$ on the SDC can be extracted. To this end, the buildings are separated into two categories: the buildings that are classified to the "S" SDC for $\theta = 0$ (i.e., MIDR ($\theta = 0$) < 0.5%), and those that are classified to the "M" SDC for $\theta = 0$ (i.e., 0.5% < MIDR ($\theta = 0$) < 1.0%). For buildings that are classified to the "S" SDC for $\theta = 0$, the problem can be defined as a two- or three-class PR problem. More specifically, the consideration of a three-class PR problem (Approach 2/Category 3S, or A2/C3S) leads to the prediction of the exact category of buildings' SDC for $\theta = \theta_{cr}$, and not only to the prediction of the change (or not) in the SDC for $\theta = \theta_{cr}$. In other words, in this case the three classes are defined by means of the following criteria: Class 1: {SDC ($\theta = 0$) = "S" → SDC ($\theta = \theta_{cr}$) = "S"}, Class 2: {SDC ($\theta = 0$) = "S" → SDC ($\theta = \theta_{cr}$) = "M"}, and Class 3: {SDC ($\theta = 0$) = "S" → SDC ($\theta = \theta_{cr}$) = "H"}. Class 1 corresponds to insignificant influence of the $\theta_{cr}$ on the SDC, whereas Classes 2 and 3 correspond to significant influence. Correspondingly, in case of the two-class PR problem (Approach 2/Category 2S, or A2/C2S), the two classes are defined by means of the following criteria: Class 1: {SDC ($\theta = 0$) = "S" → SDC ($\theta = \theta_{cr}$) = "S"}, and Class 2: {SDC ($\theta = 0$) = "S" → SDC ($\theta = \theta_{cr}$) = "M" or SDC ($\theta = \theta_{cr}$) = "H"}. Finally, in the framework of the second approach, a separate procedure is followed for buildings that are classified to the "M" SDC for $\theta = 0$ (Approach 2/Category 2M, or A2/C2M). More specifically, in this case, only two classes can be defined, i.e., Class 1: {SDC ($\theta = 0$) = "M" → SDC ($\theta = \theta_{cr}$) = "M"} and Class 2: {SDC ($\theta = 0$) = "M" → SDC ($\theta = \theta_{cr}$) = "H"}.

The two aforementioned approaches for the solution to the problem of the present study are summarized in Table 2 (where the correspondence of each class to the significance or the insignificance of the influence of $\theta_{cr}$ on the SDC is also presented).

Obviously, A1 concerns all buildings, regardless of their classification to an SDC for $\theta = 0$. Thus, this approach is applicable whether the classification of buildings to an SDC for $\theta = 0$ is known or not. On the other hand, if A2 is chosen, the problem is solved by selecting one of the categories A2/C3S or A2/C2S in combination with the category A2/C2M. In other words, the solution of the problem using A2 is achieved by the decision to formulate it using one of the following pairs (analysis types): "A2/(C3S + C2M)" or "A2/(C2S + C2M)". Thus, the solution of the problem using one of the two forms of A2 is feasible only if the SDC of buildings for $\theta = 0$ is known.

**Table 2.** Main characteristics of the approaches used for the definition of the studied problem.

| Approach/Category | Classes | Criteria of Classes/Significant Influence of $\theta_{cr}$ |
|---|---|---|
| A1 | 2 | Class 1: SDC ($\theta = \theta_{cr}$) = SDC ($\theta = 0$)/NO<br>Class 2: SDC ($\theta = \theta_{cr}$) > SDC ($\theta = 0$)/YES |
| A2/C3S | 3 | Class 1: SDC ($\theta = 0$) = "S" → SDC ($\theta = \theta_{cr}$) = "S"/NO<br>Class 2: SDC ($\theta = 0$) = "S" → SDC ($\theta = \theta_{cr}$) = "M"/YES<br>Class 3: SDC ($\theta = 0$) = "S" → SDC ($\theta = \theta_{cr}$) = "H"/YES |
| A2/C2S | 2 | Class 1: SDC ($\theta = 0$) = "S" → SDC ($\theta = \theta_{cr}$) = "S"/NO<br>Class 2: SDC ($\theta = 0$) = "S" → SDC ($\theta = \theta_{cr}$) = "M" or "H"/YES |
| A2/C2M | 2 | Class 1: SDC ($\theta = 0$) = "M" → SDC ($\theta = \theta_{cr}$) = "M"/NO<br>Class 2: SDC ($\theta = 0$) = "M" → SDC ($\theta = \theta_{cr}$)= "H"/YES |

The categories of the MLP networks that were used for each of the aforementioned alternative formulations of the problem are summarized in Table 3. It must be noted that the names of networks in Table 3 concern the type of analysis; their complete names also contain the types of buildings (i.e., BB, RIB, IIB), as presented in Section 4.

**Table 3.** Main characteristics of the types of analyses and the corresponding names of networks.

| Analysis Type | Names of Networks | Number of Classes |
|---|---|---|
| "A1" | NA1 (all buildings) | 2 |
| "A2/(C3S + C2M)" | NA2C3S (buildings classified to SDC "S" for $\theta = 0$)<br>NA2C2M (buildings classified to SDC "M" for $\theta = 0$) | 3 if SDC ($\theta = 0$) = "S"<br>2 if SDC ($\theta = 0$) = "M" |
| "A2/(C2S + C2M)" | NA2C2S (buildings classified to SDC "S" for $\theta = 0$)<br>NA2C2M (buildings classified to SDC "M" for $\theta = 0$) | 2 if SDC ($\theta = 0$) = "S"<br>2 if SDC ($\theta = 0$) = "M" |

On the basis of the selected approach (A1 or A2), the general form of the procedure used in the present study for the investigation of the significance (or the insignificance) of the influence of the critical seismic angle on RC buildings' SDC (considering that the used MLP networks are already trained) is described by means of the flowcharts shown in Figure 3.

It must be noted that when A2 is used there are two alternatives for the estimation of the SDC ($\theta = 0$): (a) with application of NTHA, or (b) using MLP networks (Figure 3b). In order to adopt the alternative (b), it is necessary for appropriately trained networks to be available (N$\theta$0 networks in Figure 3b). These networks must be trained in parallel with the networks that predict the level of influence of $\theta_{cr}$ on SDC. The procedure and the details for the configuration and the training of N$\theta$0 networks are given by Morfidis and Kostinakis [32]. However, in the present study, the source of the information about the classification of buildings in SDC for $\theta = 0$ (i.e., NTHA or N$\theta$0 networks) did not affect the procedure of investigation.

### 3.3. Selection of Ground Motion, RC Buildings, and the Training Dataset Generation

The training dataset generation using numerical procedures initially requires the proper selection of RC buildings and ground motion records. After these selections, as is well known, the training dataset is generated through a series of NTHA of each of the selected buildings for all of the selected ground motion records. The whole procedure for the generation of a training dataset using a numerical approach is fully described in [32]. This type of procedure was also used in the present study; however, a significant differ-

entiation was necessary herein. More specifically, in contrast to the procedure described in [32], each of the selected RC buildings was analyzed for all of the selected ground motion records considering not only the seismic angle θ = 0, but also seismic angles between 0 and 355 degrees (θ = 0°, 5°, 10°, . . . ., 355°), as illustrated in Figure 4 (for more details, see also [6]). Therefore, for each of the selected RC buildings and each of the selected ground motion records, 72 (= 360/5) NTHAs were performed using the computer program Ruaumoko [55]. The seismic angle for which the MIDR value is maximized is the $\theta_{cr}$. Then, based on the corresponding MIDR values, the classification of each of the selected RC buildings in the three predefined SDCs of Table 1 for the seismic angles θ = 0 (SDC (θ = 0)) and θ = $\theta_{cr}$ (SDC (θ = $\theta_{cr}$)) is accomplished and stored using the procedure shown in Figure 4b. Finally, the sets of target vectors in forms compatible with A1 and A2 (Tables 2 and 3) are formed.

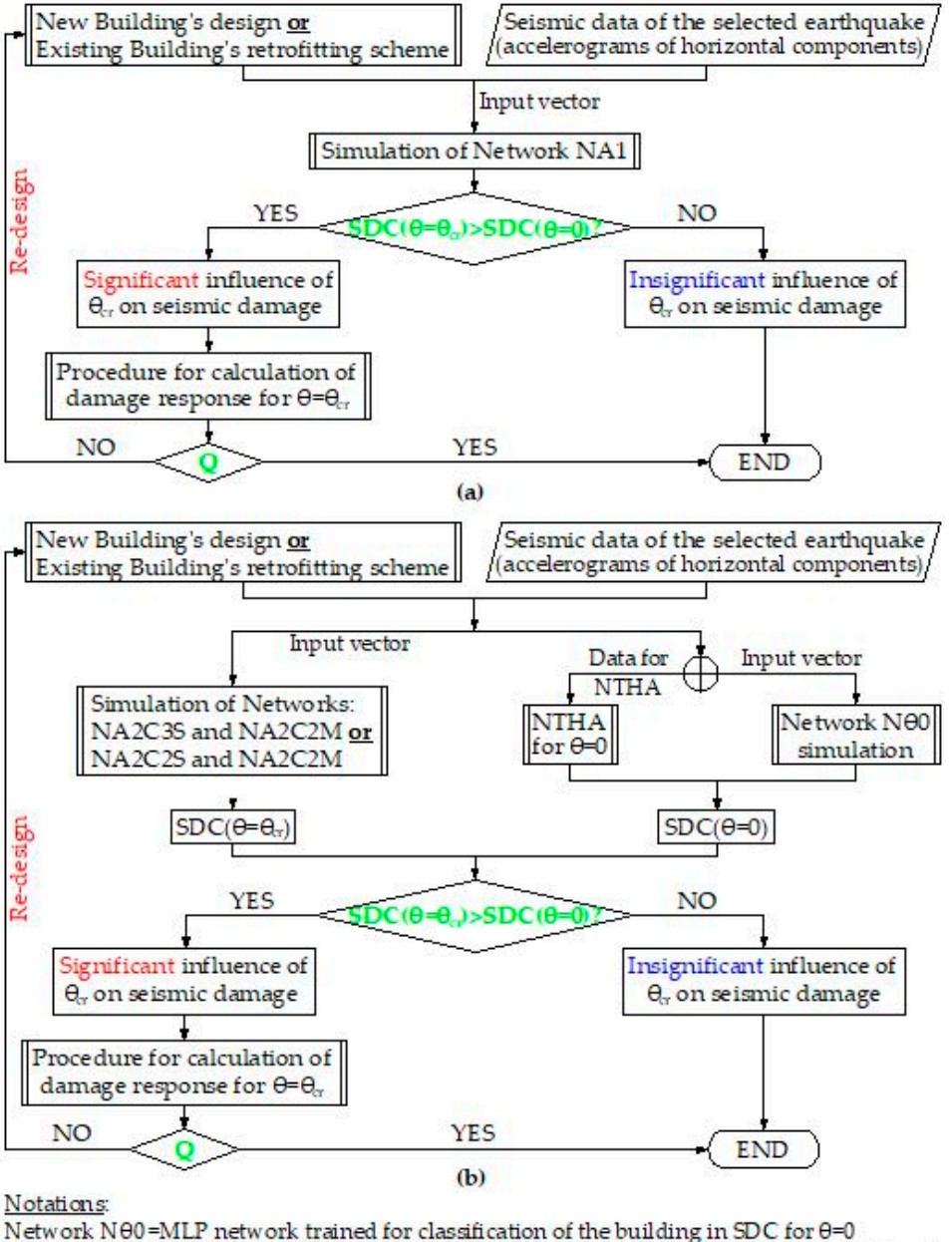

**Figure 3.** General flowcharts of the procedure for the study of the critical seismic angle's influence on the SDC of RC buildings: (**a**) A1; (**b**) A2.

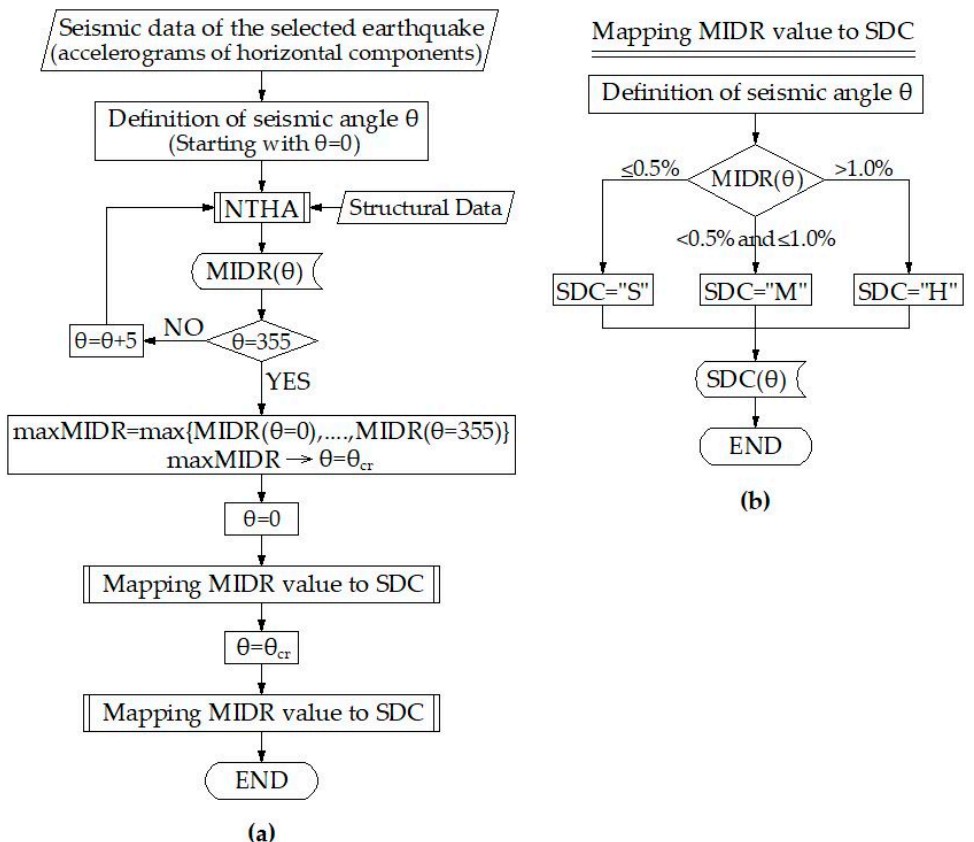

**Figure 4.** Flowchart of the procedure used for the generation of the training datasets (**a**,**b**).

As regards the selected RC buildings, 30 different types in terms of their structural systems were configured, modelled, and designed according to the provisions of EN1992-1-1 [56] and EN1998-1 [16]. These buildings were regular in elevation according to EN1998-1. Due to the fact that the selected buildings were used in previous published research, more details and full description of their configuration and modelling (for design purposes and for the NTHA) can be found in [32,33]. However, the most significant points regarding the modelling assumptions of the selected RC buildings, as well as their geometric characteristics, are presented in Appendix A. It must be stressed herein that for each of the 30 selected RC buildings, 3 different versions were considered as regards their masonry infills (see also [33]): (1) buildings without masonry infills (Bare Buildings; BB), (2) buildings with masonry infills in all stories (Regularly Infilled Buildings; RIB), and (3) buildings with masonry infills in all stories except for the ground story (Irregularly Infilled Buildings; IIB). For each of these three versions of the buildings, separate training datasets were generated according to A1 and A2 (Tables 2 and 3). Thus, all analyses presented in this paper (Section 4) were performed separately for each of the three aforementioned versions of the selected buildings.

Finally, as regards the selected seismic ground motions, the 65 recorded ground motions that are presented in [32] were used in the present study (Appendix B). It must be noted that these ground motions were imported to the models as recorded, and then rotated with respect to the axes of buildings. For the generation of each of the three training datasets, 140,400 (= 30 buildings × 65 ground motions × 72 seismic angles) NTHAs were performed. This means that the procedure described in Figure 4 was performed 140,400/72 = 1950 times for each of the three versions of the 30 selected RC buildings. The number of the target vectors and their form (which corresponds to the networks that are presented in Table 3) for each of the three versions of the selected RC buildings are given in Table 4.

**Table 4.** Number of samples and the form of target vectors for each of the networks and the versions of the selected RC buildings.

| Network | Number of Samples | Form of Target Vectors (TV) |
|---------|-------------------|------------------------------|
| NA1 | BB: 1006 (=1950–944 *), RIB: 1539 (=1950–411 *), IIB: 1052 (=1950–898 *) | TV = [1 0] $^{\text{T}}$ (Class 1) <br> TV = [0 1] $^{\text{T}}$ (Class 2) |
| NA2C3S | BB: 563 (=1950–443–944 *), RIB: 1075 (=1950–464–411 *), IIB: 673 (=1950–379–898 *) | TV = [1 0 0] $^{\text{T}}$ (Class 1) <br> TV = [0 1 0] $^{\text{T}}$ (Class 2) <br> TV = [0 0 1] $^{\text{T}}$ (Class 3) |
| NA2C2S | BB: 563, RIB: 1075, IIB: 673 | TV = [1 0] $^{\text{T}}$ (Class 1) <br> TV = [0 1] $^{\text{T}}$ (Class 2) |
| NA2C2M | BB: 443 (=1950–563–944 *), RIB: 464 (=1950–1075–411 *), IIB: 379 (=1950–673–898 *) | TV = [1 0] $^{\text{T}}$ (Class 1) <br> TV = [0 1] $^{\text{T}}$ (Class 2) |

* Number of samples that are classified to the SDC "H" for $\theta = 0$.

### 3.4. Selection of Parameters for the Input Vectors

A very important part of the present study was the selection of parameters for the input vectors of the used MLP networks. Generally, in problems that concern the prediction of seismic damage to buildings, the input vectors of the MLP networks must contain seismic and structural parameters (see, e.g., [23,28,29,57,58]).

As regards the seismic parameters, previous studies indicate that the well-documented seismic parameters that are defined and used in the specific literature (see, e.g., [59]) lead to reliable results. Taking into consideration the fact that in analyses of 3D RC buildings the vertical component of earthquake records is usually neglected (except for the case of structures with special characteristics, which are not the subject of the current study), the input vectors must contain values of seismic parameters that are extracted from a selected combination rule of their values along the two horizontal components of the excitation (i.e., their geometric mean value, e.g., [60]). However, due to the fact that the target of the present study is the investigation of the influence of $\theta_{\text{cr}}$ on the SDL of RC buildings, it is more rational to select the ratios of their values along the two horizontal components of the seismic excitation as seismic parameters in the input vectors, rather than their values extracted by a combination rule. Thus, the ratios of the minimum to the maximum value of the selected seismic parameters were used in input vectors for MLP networks, as is presented by the general Equation (2).

$$\text{SIP} = \frac{\min\{\text{SIP(dir1)}; \text{SIP(dir2)}\}}{\max\{\text{SIP(dir1)}; \text{SIP(dir2)}\}} \tag{2}$$

where:

SIP is the value of the s̲eismic i̲nput p̲arameter that is introduced to the input vector;
SIP (dir1) is the value of the s̲eismic i̲nput p̲arameter that is extracted from the accelerogram that corresponds to horizontal direction $\overline{1}$ of the seismic excitation;
SIP (dir2) is the value of the s̲eismic i̲nput p̲arameter that is extracted from the accelerogram that corresponds to horizontal direction $\overline{2}$ of the seismic excitation.

It must be noted that directions 1 and 2 correspond to the horizontal axes along which the seismic excitations' accelerograms are recorded (see Figure 1).

The selection of seismic parameters was based on the corresponding selection that was made in previous published studies (see, e.g., [32]), but it was necessary to make some changes herein. More specifically, the seismic parameters that are based on the earthquakes' duration—namely, the uniform duration, the bracketed duration, and the significant duration (see, e.g., [59])—were excluded due to the fact that their values for one

or both of the horizontal directions for some of the 65 earthquakes used for the generation of the training dataset were equal to zero. This led to problems in the application of Equation (2) (0/0 ratios); thus, 12 selected seismic parameters were introduced in the input vectors in the present study, and are summarized in Table 5.

**Table 5.** The selected seismic parameters for the input vectors of MLP networks.

| | | | |
|---|---|---|---|
| 1 | Peak ground acceleration (PGA) | 7 | Acceleration spectrum intensity (ASI) |
| 2 | Peak ground velocity (PGV) | 8 | Cumulative absolute velocity (CAV) |
| 3 | Specific energy density (SED) | 9 | Peak ground displacement (PGD) |
| 4 | Arias intensity ($I_a$) | 10 | Effective peak acceleration (EPA) |
| 5 | Predominant period (PP) | 11 | Sustained maximum acceleration (SMA) |
| 6 | Housner intensity (HI) | 12 | Sustained maximum velocity (SMV) |

Many structural parameters can be used for the description of the seismic response of RC buildings. These parameters generally quantify the stiffness and the strength of buildings under seismic loads. The nature of the present study, which mainly concerns a geometric problem, along with the above-described rationale of selection of the seismic input parameters (Equation (2)), led to the selection of the following three <u>st</u>ructural <u>in</u>put <u>p</u>arameters (StIPs):

(a)  The ratio of the minimum to the maximum value of uncoupled fundamental natural periods $T_{1,X}$, $T_{1,Y}$ of buildings for pure vibration along their structural axes X and Y, respectively (see Figure 5).

$$\text{StIP1} = \min\{T_{1,X}; T_{1,Y}\}/\max\{T_{1,X}; T_{1,Y}\} \tag{3}$$

This parameter is an index of the relative horizontal stiffness of buildings along their two orthogonal structural axes. The values of the uncoupled fundamental natural periods are connected to the initial choice of structural axes, which are generally defined in different ways, as stated above. The choice of fundamental natural periods was made in order to define a "metric" by which to measure the ratio of the horizontal stiffness of buildings along two perpendicular axes, which is not influenced by "coupling" effects. It must also be noted that the structural axes can have any orientation, but this fact does not affect the whole procedure, since the uncoupled fundamental natural periods can be defined in any system of two orthogonal axes;

(b)  The ratio of the buildings' total height $H_{tot}$ to the square root of the sum of squares of the horizontal dimensions $L_X$ and $L_Y$ of their plans along the structural axes X and Y, respectively (see Figure 5).

$$\text{StIP2} = H_{tot}/\sqrt{L_X^2 + L_Y^2} \tag{4}$$

This parameter expresses the slenderness of buildings, and plays a significant role in their seismic response, because it gives an additional index of the horizontal stiffness. It must be noted that this parameter can be defined not only in cases of buildings with rectangular plans, but also in any case using equivalent horizontal dimensions along the structural axes X and Y (see Figure 5).

(c)  The ratio of the structural eccentricity (i.e., the distance between the mass center (MC) and the stiffness center (SC) of stories) $e_0$ to the dimensions of the plan of the building parallel to it (see Figure 5).

$$\text{StIP3} = e_0/L_{e0} \tag{5}$$

This parameter indicates the degree of eccentricity of the forces induced by seismic excitations; it is well documented that this degree significantly affects the level of seismic damage.

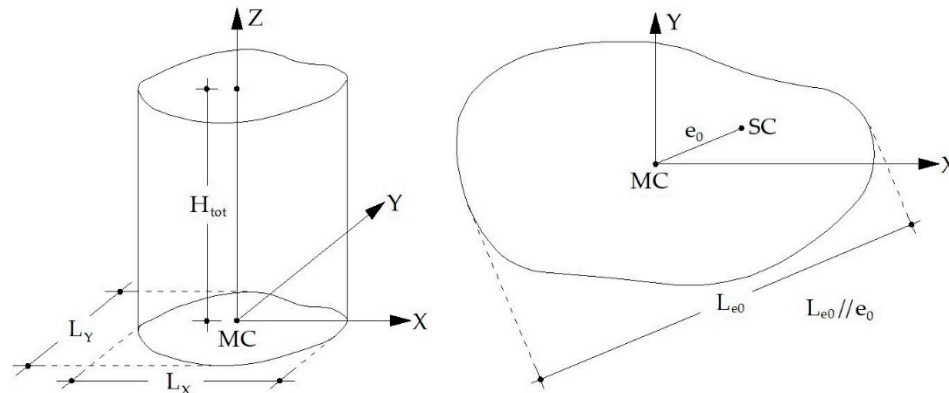

**Figure 5.** Definition of the selected structural parameters for the input vectors of MLP networks.

Therefore, in the present study, 15 input parameters (12 seismic parameters and 3 structural parameters) were used as elements of the input vectors of MLP networks. The general form of these input vectors (dimensions $15 \times 1$) is given by Equation (6).

$$
\begin{aligned}
\mathbf{x} &= \begin{bmatrix} \mathbf{x}_{\text{seism}} & \mathbf{x}_{\text{struct}} \end{bmatrix}^T \\
\mathbf{x}_{\text{seism}} &= \begin{bmatrix} \dfrac{\text{PGA}_{\min}}{\text{PGA}_{\max}} & \dfrac{\text{PGV}_{\min}}{\text{PGV}_{\max}} & \dfrac{\text{PGD}_{\min}}{\text{PGD}_{\max}} & \dfrac{\text{I}_{a,\min}}{\text{I}_{a,\max}} & \dfrac{\text{SED}_{\min}}{\text{SED}_{\max}} & \dfrac{\text{CAV}_{\min}}{\text{CAV}_{\max}} \\ \dfrac{\text{ASI}_{\min}}{\text{ASI}_{\max}} & \dfrac{\text{HI}_{\min}}{\text{HI}_{\max}} & \dfrac{\text{EPA}_{\min}}{\text{EPA}_{\max}} & \dfrac{\text{SMA}_{\min}}{\text{SMA}_{\max}} & \dfrac{\text{PP}_{\min}}{\text{PP}_{\max}} & \dfrac{\text{SMV}_{\min}}{\text{SMV}_{\max}} \end{bmatrix}^T \\
\mathbf{x}_{\text{struct}} &= \begin{bmatrix} \dfrac{\min\{T_{1,X};T_{1,Y}\}}{\max\{T_{1,X};T_{1,Y}\}} & \dfrac{H_{\text{tot}}}{\sqrt{L_X^2 + L_Y^2}} & \dfrac{e_0}{L_{e0}} \end{bmatrix}^T
\end{aligned}
\tag{6}
$$

### 3.5. Parametric Investigation for the Optimal Configuration of the Used MLP Networks

It is well known that the performance of the MLP networks is strongly affected by their configuration, i.e., the selection of the parameters that constitute their structure (Figure 2). These parameters are the number of hidden layers, the neurons in each one of the hidden layers, and the activation functions of the neurons (see e.g., [61]). Furthermore, additional choices affect the solution of problems by means of MLP networks, including the normalization functions of the input and output vectors, the procedure for the partitioning of the generated dataset in training, validation, and testing subsets, the performance evaluation parameters, and the training algorithms.

In the present study, the choices of the aforementioned parameters are summarized in Figure 6. More details about the procedure of the parametric investigation for the optimal selection of the parameters that affect the performance of MLP networks are presented in [32]. The results of this investigation are demonstrated and discussed in Section 4.

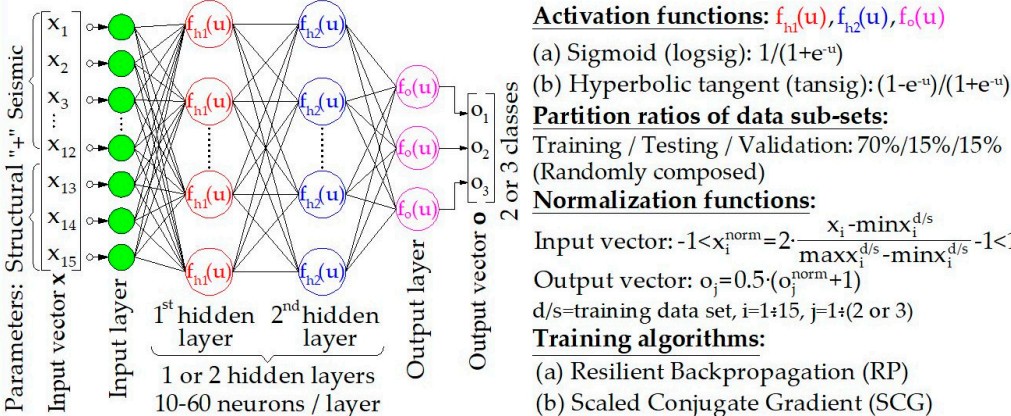

**Figure 6.** The general form of the used networks and the investigated configuration parameters.

## 4. Presentation and Evaluation of the Results of the Training Procedures

In this section, the results extracted from the procedures described in Section 3 are presented and evaluated. The basic performance evaluation metric for the studied MLP networks is the overall accuracy (OA) index. The OA is the ratio of the samples of a dataset that are classified to the correct class to the total number of samples of that dataset. This index is usually a part of the confusion matrices (CMs) (see, e.g., [34,62]). Additionally, other parameters that are defined in CMs are also used if the OA comparisons do not lead to clear conclusions; thus, the entire CMs are also presented. As presented in Figure 6, the generated dataset (Section 3.3) is partitioned (for the purposes of the training procedures) into three subsets: the training subset, the validation subset, and the testing subset. The results of the investigations for the optimal configuration of the MLP networks using the selected metrics (the OA index and the other indices of CMs) are presented for the testing subset, the training subset, and the total dataset in any case. However, the performance evaluation using the selected metrics is based on the testing subsets, because these subsets extract reliable estimations about the generalization ability of the MLP networks (i.e., the contribute to checking for the existence (or non-existence) of overfitting during the training procedure (see, e.g., [63,64])).

As noted in Section 3.2, the two approaches to the solution of the problem (A1, A2) lead to three different analysis types (Table 3). The two different form of approach A2 (i.e., "A2/(C3S + C2M)" and "A2/(C2S + C2M)") differ only in the definition of the number of SDCs ($\theta = \theta_{cr}$) for buildings that are classified to the "S" class for $\theta = 0$ (three SDCs ($\theta = \theta_{cr}$) in the case of C3S and two SDCs ($\theta = \theta_{cr}$) in the case of C2S). On the other hand, the number of classes for buildings that are classified to the SDC "M" for $\theta = 0$ is two in all cases (Table 3). Therefore, in order to compare the A1 and A2, the comparison of the two forms of A2 must first be performed. More specifically, the categories C3S and C2S must first be compared, and then the more reliable of the two—in conjunction with C2M—can be compared with A1.

The whole procedure for the evaluation of A1 and A2 was performed separately for the three different versions of the selected RC buildings (i.e., BB, RIB, and IIB); thus, the presentation of the results follows this rationale. Finally, it must be noted that the presentation of the results in this section is separated into three parts: In the first part (Section 4.1), the results of the investigation for the optimal configuration of the MLP networks for A1 are presented and discussed. The second part (Section 4.2.) concerns the corresponding procedure for A2. Finally, in the third part (Section 4.3), the results of the comparison between the optimally configured MLP networks of the two approaches are discussed.

### 4.1. Optimal Configuration of the MLP Networks Used for the Implementation of A1

In Figure 7, the values of the OA index for the three versions of the selected RC buildings are presented. More specifically, this figure contains the maximum values of the OA index extracted by the optimally configured MLP networks with one or two hidden layers trained using the resilient backpropagation (RP) algorithm (see, e.g., [65]) and the scaled conjugate gradient (SCG) algorithm (see, e.g., [66]). In Table 6, the corresponding configuration parameters of the optimal MLP networks (i.e., number of hidden layers, number of neurons/hidden layer, and activation functions of neurons) that extract the OA values of Figure 7 are summarized.

From the study of Figure 7, the following main conclusions can be drawn:

- The networks with two hidden layers extract greater OA values than the networks with one hidden layer; however, the differences between them are not significant. More specifically, in the case of training using the RP algorithm, these differences fluctuate between 3.1 and 6.7% for BB, 2.2 and 6.7% for RIB, and 5.3 and 9.5% for IIB. The corresponding fluctuations in the case of training using the SCG algorithm are 2.1–3.1% (BB), 2.7–3.8% (RIB), and 3.2–3.8% (IIB);

- As regards the evaluation of the used training algorithms for the networks with two hidden layers, the RP algorithm is in all cases more effective than the SCG algorithm. However, the extracted OA index values are generally acceptable regardless of the used training algorithm. More specifically, the RP algorithm extracts OA values higher than 82% (in the case of BB and IIB buildings, the values are almost equal to 90%), whereas the corresponding values extracted by the SCG algorithm are slightly lower.

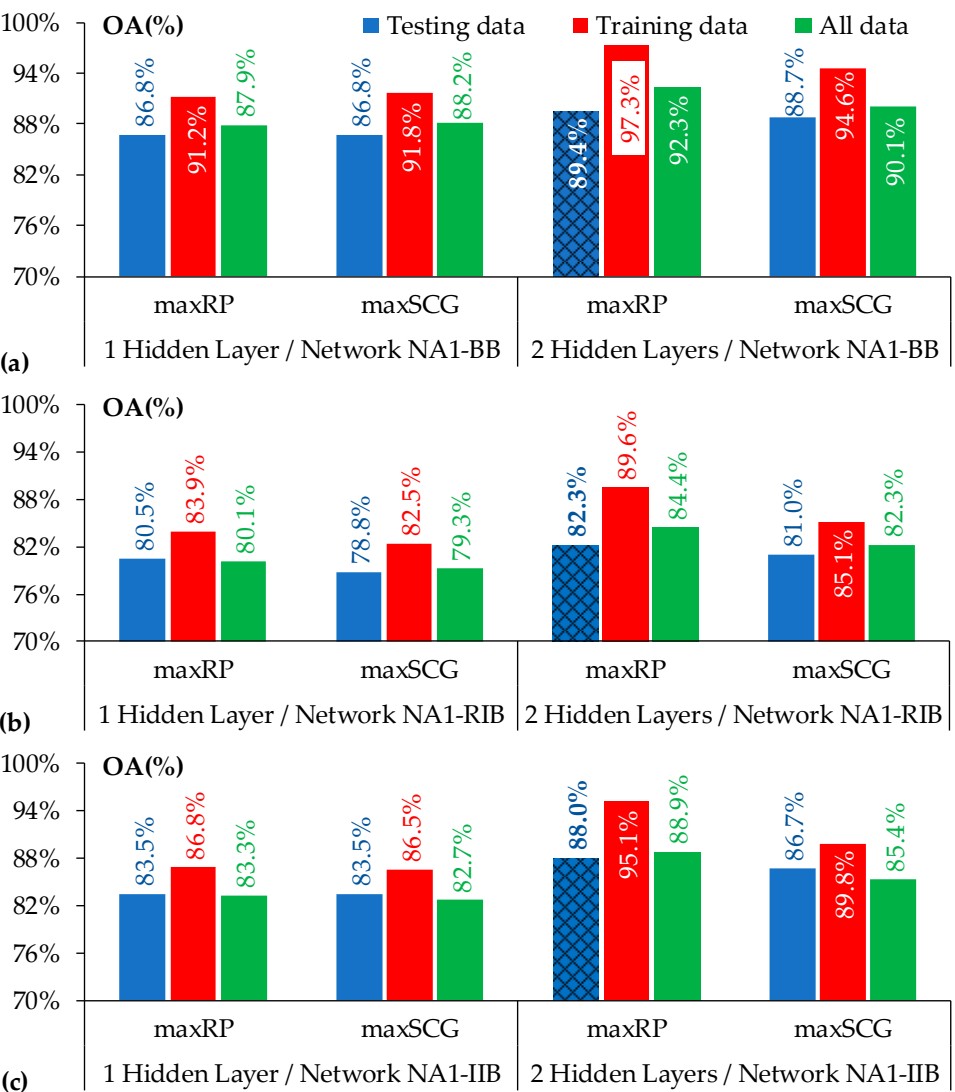

**Figure 7.** Maximum OA index values extracted by NA1 networks with 1 or 2 hidden layers: (**a**) BB; (**b**) RIB; (**c**) IIB.

The basic conclusion that can be extracted from the study of Table 6 is that, with only one exception, the configuration of all optimal networks includes the tansig activation function in the output layer.

Due to the fact that the above-mentioned differences between the maximum values of the OA index extracted by the networks with one or two hidden layers are not significant, a further examination of their performance using the other parameters defined through the CMs is required. Thus, in Figure 8, the CMs of the optimally configured networks with one and two hidden layers, which correspond to the testing subsets for the three versions of the selected RC buildings, are presented.

**Table 6.** The parameters of the optimally configured networks used for the implementation of A1.

| Version of Buildings | | BB | | | | RIB | | | | IIB | | | |
|---|---|---|---|---|---|---|---|---|---|---|---|---|---|
| Training Algorithm | | RP | | SCG | | RP | | SCG | | RP | | SCG | |
| Number of HL | | 1 | 2 | 1 | 2 | 1 | 2 | 1 | 2 | 1 | 2 | 1 | 2 |
| Training data-set | Activation functions | T/T | T/T/T | T/T | T/L/T | T/T | T/T/T | L/T | T/T/T | T/T | T/T/T | T/T | T/T/T |
| | Neurons/HL | 60 | 60/50 | 52 | 48/52 | 46 | 52/52 | 30 | 52/48 | 56 | 50/46 | 46 | 58/46 |
| Testing dataset | Activation functions | T/T | **L/L/T** | L/T | T/T/T | T/T | **L/L/T** | L/T | T/L/T | L/T | **T/T/T** | T/T | L/T/L |
| | Neurons/HL | 30 | **36/34** | 14 | 28/50 | 30 | **28/14** | 52 | 58/52 | 24 | **20/10** | 14 | 10/12 |
| Total dataset | Activation functions | L/T | T/T/T | T/T | T/L/T | T/T | T/T/T | T/T | T/T/T | T/T | T/T/T | T/T | T/T/T |
| | Neurons/HL | 30 | 60/50 | 58 | 48/52 | 40 | 52/52 | 60 | 50/26 | 60 | 50/60 | 40 | 26/28 |

**Notation**—T: tansig, L: logsig, HL: hidden layer.

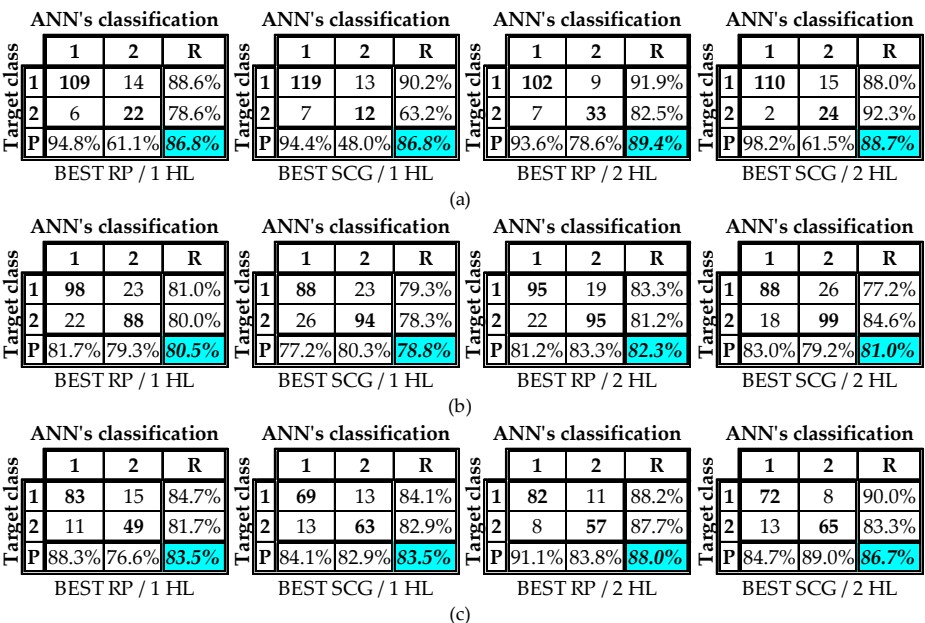

(a)

(b)

(c)

**Figure 8.** CMs according to the testing subsets extracted by the optimally configured NA1 networks: (**a**) BB; (**b**) RIB; (**c**) IIB.

From the study of Figure 8, it is obvious that the networks with two hidden layers are more efficient than the networks with one hidden layer—not only on the basis of the OA index values, but also according to the other two indices that are defined in the CMs, namely, the recall (R) index and the precision (P) index. More specifically, the networks that are the most efficient according to the OA index (Figure 7) also generally have higher R-index and P-index values for the three versions of the studied RC buildings (i.e., BB, RIB, and IIB). This means that the optimally configured networks with two hidden layers efficiently classify the studied samples in each of the two SDCs. Therefore, the high OA values of Figure 7 are extracted due to correct classifications to both of the two predefined SDCs, and not to only one of them.

After the above-presented study of the results extracted from the trained networks used for the solution of the problem according to A1, it can be concluded that the most efficient (optimally configured) networks on the basis of the testing subset are those that are summarized in Table 7.

**Table 7.** Configuration parameters and names of the optimal networks used for A1.

| Version of Buildings | BB | RIB | IIB |
|---|---|---|---|
| Training algorithm | RP | RP | RP |
| Number of HLs | 2 | 2 | 2 |
| Activation functions | L/L/T | L/L/T | T/T/T |
| Neurons/HL | 36/34 | 28/14 | 20/10 |
| Name of network | NA1-BB | NA1-RIB | NA1-IIB |

**Notation**—T: tansig; L: logsig; HL: hidden layer.

### 4.2. Optimal Configuration of the MLP Networks Used for the Implementation of A2

As mentioned above (and in Table 3), A2 is defined using two alternative types that differ in the number of predefined SDCs for the buildings that are classified to the SDC "S" for θ = 0 (categories C3S and C2S). On the other hand, only one category for classification of buildings that are classified to the SDC "M" for θ = 0 is defined (category C2M). Thus, in order to compare the two types of A2 (i.e., "A2/(C3S + C2M)" and "A2/(C2S + C2M)" in Table 3), a comparison between the categories C3S (NA2C3S networks) and C2S (NA2C2S networks) must be performed.

4.2.1. Comparative Evaluation of the Optimally Configured NA2C3S and NA2C2S Networks

In Figure 9, the OA index values extracted by the optimally configured networks NA2C3S and NA2C2S for the three versions of the selected RC buildings are presented.

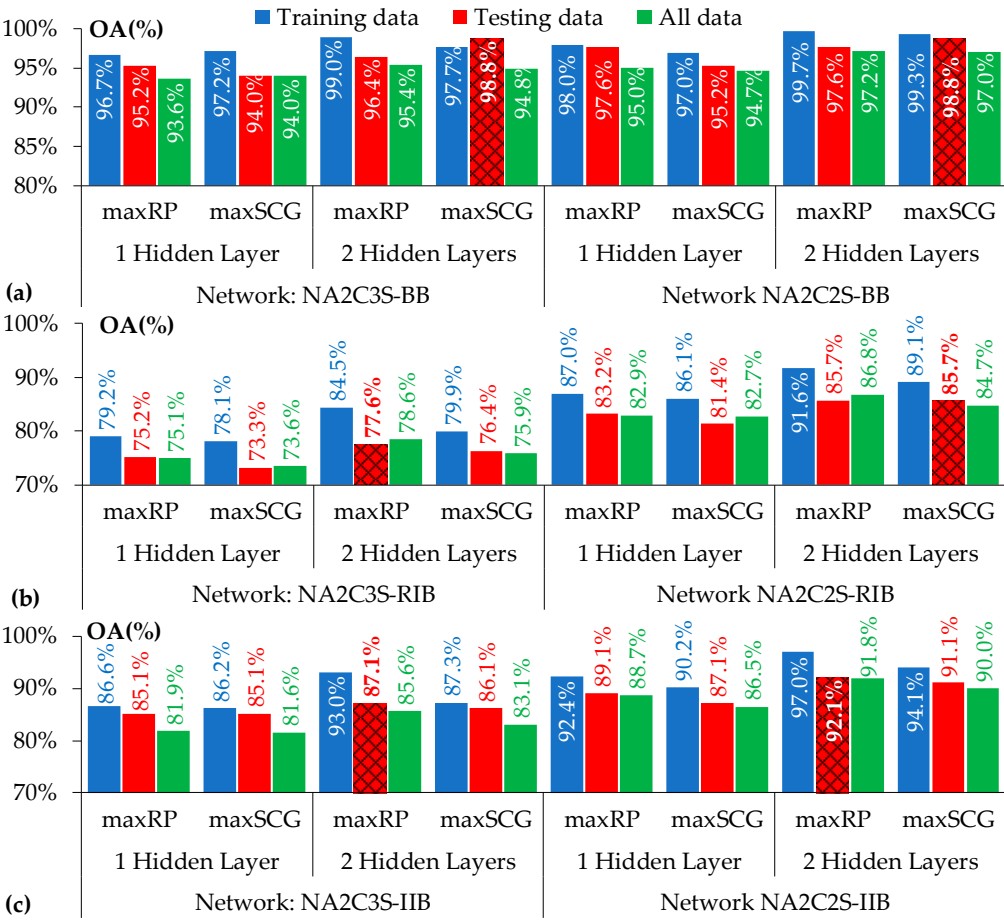

**Figure 9.** Maximum OA index values extracted by NA2C3S and NA2C2S networks with 1 or 2 hidden layers: (**a**) BB; (**b**) RIB; (**c**) IIB.

From the study of Figure 9, the following main conclusions can be extracted:

- In general, C2S extracts more reliable predictions than C3S. This conclusion is valid for the three versions of the RC buildings. However, the differences between the two categories are not significant, especially in the case of BB (maximum difference = 2.44%). For the other two versions of buildings, the differences are greater, but still not significant (the maximum differences are 11% for RIB and 7.8% for IIB);
- According to the testing subsets, which are generally used in this paper for comparisons, no clear conclusion can be reached as regards the most efficient training algorithm. In the case of C3S, the SCG algorithm is more efficient than the RP only for BB. On the other hand, in the case of C2S, the RP algorithm is more efficient than the SCG only for IIB;
- The addition of the second hidden layer improves the OA index values, but not significantly as regards the comparisons using the testing subsets. For BB, the differences in maximum OA index values between the networks with one and two hidden layers are 4.8% in the case of C3S and 3.6% in the case of C2S; for RIB, the corresponding differences are 3.2% in the case of C3S and 5.1% in the case of C2S, whereas for IIB they are 2.3% in the case of C3S and 3.2% in the case of C2S.

The results of the above analysis lead to the requirement of additional evaluation using the other metrics defined in the CMs for the extraction of a more robust conclusion as regards the most efficient category between C3S and C2S. Thus, in Figure 10, the CMs of the best configured networks of Figure 9 (i.e., the networks that extract the maximum OA index values presented in Figure 9) are illustrated.

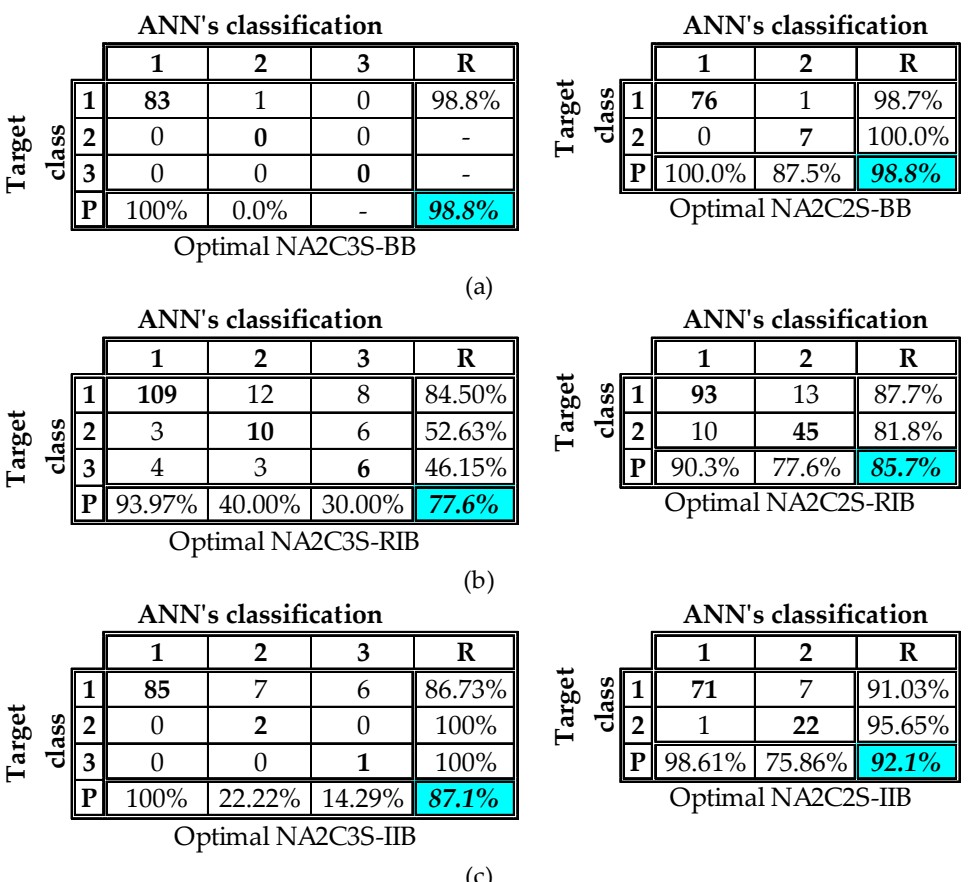

**Figure 10.** CMs according to the testing subsets extracted by the optimally configured NA2C3S and NA2C2S networks: (**a**) BB; (**b**) RIB; (**c**) IIB.

The configurations of CMs shown in Figure 10 indicate that the performance of NA2C2S networks is generally very efficient regardless of the version of RC buildings (i.e., BB, RIB, or IIB). Indeed, for all versions of all buildings, the values of the R-index and the P-index are greater than 75.8%; in addition, the vast majority of the values of these indices are greater than 85%. Conversely, as regards the NA2C3S networks, the corresponding level of efficiency for all versions of the RC buildings is not clear. In the case of BB, the values of the R- and P-factors are extremely high, but the configuration of the corresponding testing subset contains only samples that are classified to the SDC "S" for θ = 0. Therefore, no conclusion can be extracted as to the ability of the network to correctly classify the samples to other SDCs. After an extended investigation for other configurations of testing subsets (it is reminded herein that in any training procedure the subsets are formed randomly) containing all SDCs (θ = 0), it was found that the corresponding OA was smaller than 98.8%. Thus, in any case, the NA2C3S networks cannot be characterized as more effective than NA2C2S for BB. As regards RIB, is it obvious that the values of the R-index and the P-index cannot considered acceptable. Finally, the same holds for the R-index values of the CM of RIB. For the above-mentioned reasons, the C2S (NA2C2S networks) is obviously more efficient than the C3S (NA2C3S networks).

### 4.2.2. Optimal Configuration of the NA2C2M Networks

Section 4.2.1 shows that the optimally configured NA2C2S networks are more efficient than the NA2C3S ones. In this section, the results of the procedure for the investigation of the optimally configured NA2C2M networks are presented.

In Figure 11, the OA index values extracted by the optimally configured NA2C2M networks for the three versions of the selected RC buildings are summarized. From the study of Figure 11, it can be concluded that—as in the cases examined above—the networks with two hidden layers extract higher OA values than the networks with one hidden layer. In addition, these OA values are higher than 90% for all of the three versions of RC buildings. As regards the most efficient training algorithms, no general conclusions can be extracted, since for the BB and RIB the SCG algorithm is more efficient than the RP algorithm, whereas for IIB the opposite is true. However, these differences between the OA values extracted using the two algorithms are not significant—especially for the testing subsets (less than 3.3%).

In order to more robustly confirm the advantage of networks with two hidden layers, the CMs corresponding to the optimally configured networks with one and two hidden layers (Figure 11), on the basis of the testing subsets, are presented in Figure 12.

The study of Figure 12 confirms (on the basis of the R-index and P-index) the conclusion that the networks with two hidden layers are more efficient than the networks with one hidden layer. Indeed, the vast majority of the R-index and P-index values of CMs that correspond to networks with two hidden layers are greater than 90%. Conversely, the vast majority of the corresponding values of the CMs of networks with one hidden layer are lower than 90%.

Closing the presentation of the results of the investigation for the optimal configuration of MLP networks used for the implementation of A2, the parameters of the optimally configured NA2C3S, NA2C2S, and NA2C2M networks are summarized in Table 8.

### 4.3. Comparison of the Efficiency of the Optimally Configured Networks for A1 and A2

In this subsection, the results of the comparison between the optimally configured networks used for the implementation of A1 and A2 are presented and evaluated. More specifically, the networks described in Table 7 (A1) and Table 8 (A2) are compared using the percentages of the correct classifications (OA index values) extracted for the samples of the testing subsets. It should be noted here that, as regards A2, the form "A2/(C2S + C2M)" is more efficient than the form "A2/(C3S + C2M)," as shown in Section 4.2. Therefore, the results extracted by the networks NA1-BB, NA1-RIB, and NA1-IIB (Table 7) are compared to those extracted by the networks NA2C2S-BB/NA2C2M-BB, NA2C2S-RIB/NA2C2M-RIB,

and NA2C2S-IIB/NA2C2M-IIB (Table 8), respectively. Thus, the OA values extracted by the above-mentioned networks are summarized in Figure 13.

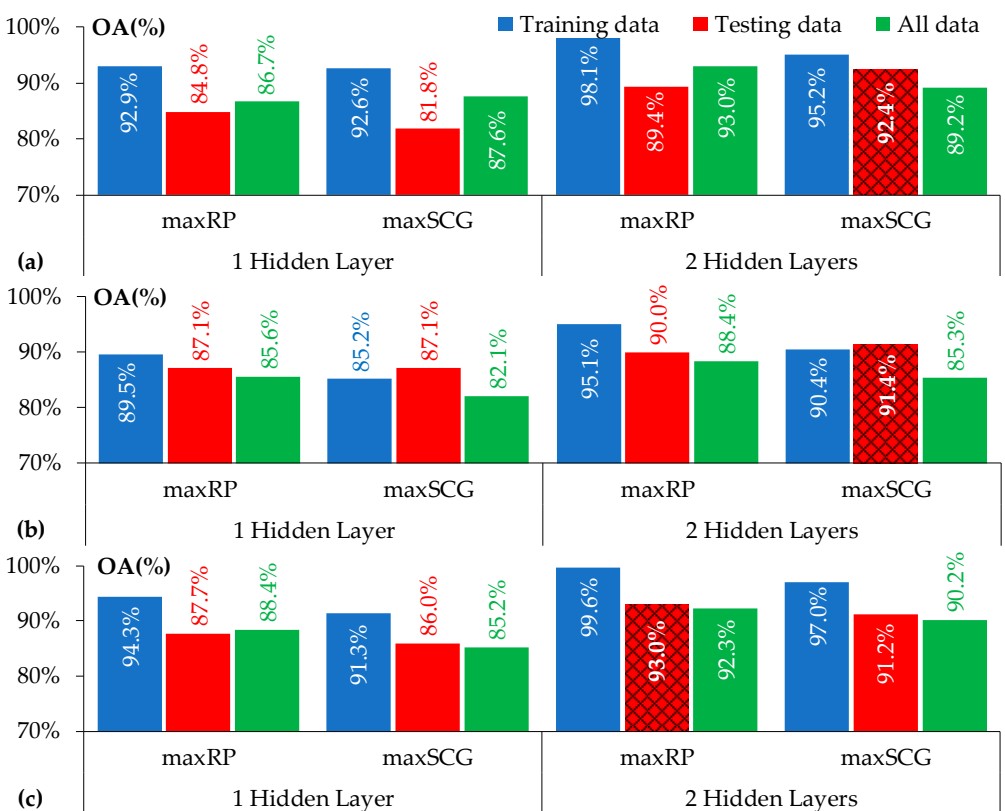

**Figure 11.** Maximum OA index values extracted by NA2C2M networks with 1 or 2 hidden layers: (**a**) BB; (**b**) RIB; (**c**) IIB.

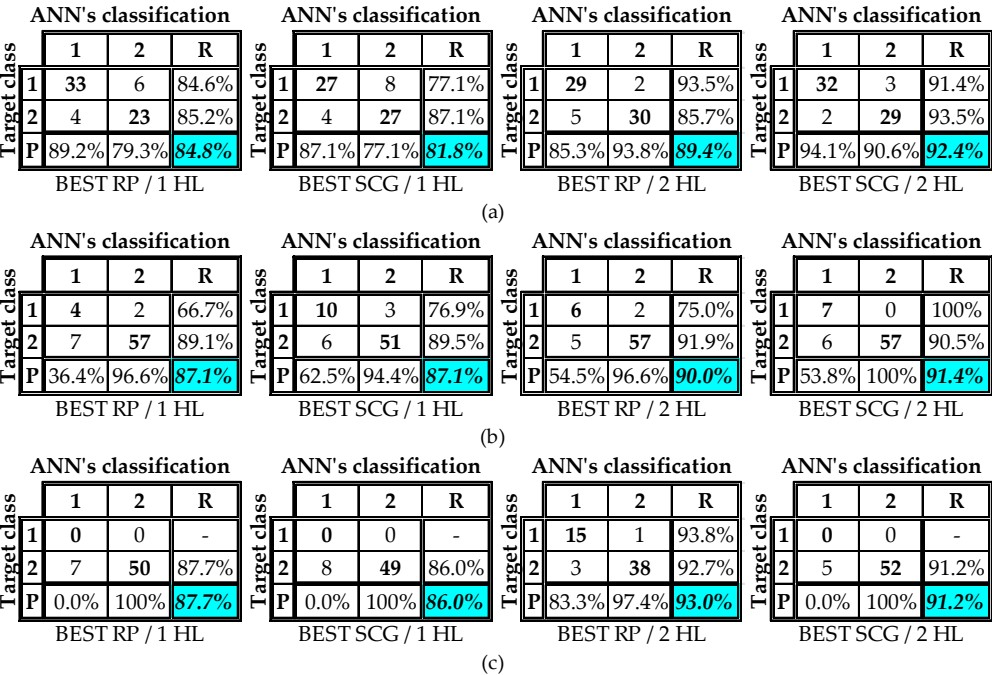

**Figure 12.** CMs according to the testing subsets extracted by the optimally configured NA2C2M networks: (**a**) BB; (**b**) RIB; (**c**) IIB.

**Table 8.** Configuration parameters and names of the optimal networks used for the implementation of A2.

| Category | A2/C3S | | | A2/C2S | | | A2/C2M | | |
|---|---|---|---|---|---|---|---|---|---|
| Version of Buildings | **BB** | **RIB** | **IIB** | **BB** | **RIB** | **IIB** | **BB** | **RIB** | **IIB** |
| Training algorithm | SCG | RP | RP | SCG | SCG | RP | SCG | SCG | RP |
| Number of HLs | 2 | 2 | 2 | 2 | 2 | 2 | 2 | 2 | 2 |
| Activation functions | T/L/L | L/L/T | L/L/T | T/T/T | T/L/T | L/T/T | L/T/T | T/T/T | T/T/T |
| Neurons/HL | 30/50 | 16/52 | 44/26 | 36/46 | 50/36 | 46/16 | 56/38 | 48/22 | 20/24 |
| Name of network | NA2C3S-BB | NA2C3S-RIB | NA2C3S-IIB | NA2C2S-BB | NA2C2S-RIB | NA2C2S-IIB | NA2C2M-BB | NA2C2M-RIB | NA2C2M-IIB |

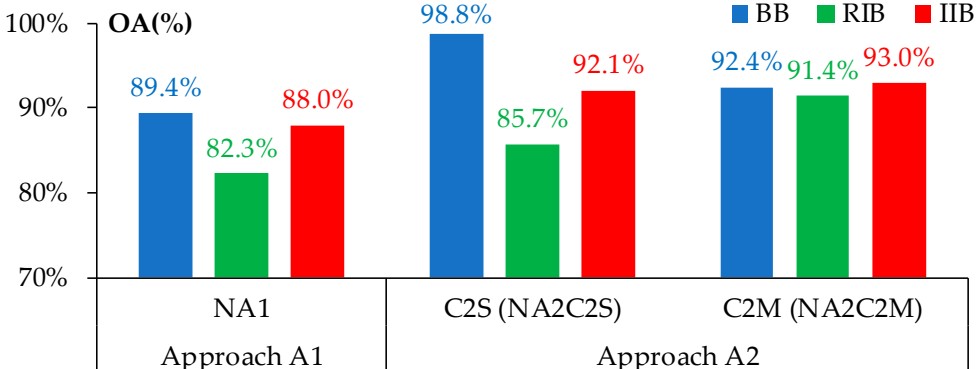

**Figure 13.** OA values extracted by the optimally configured networks for A1 and A2.

The results presented in Figure 13 indicate that A2 is clearly more efficient than A1 according to the OA index values for each of the three versions of the selected RC buildings. However, the efficiency of A1 is not unacceptable, since the OA index of the corresponding classifications is greater than 82% for RIB, whereas it is close to 90% for BB and IIB. Certainly, the vast majority of the OA values extracted by means of A2 are greater than 90%, but it must be noted that the implementation of A2 requires the knowledge of the classification of buildings to an SDC for θ = 0. This means that NTHA or the simulation of properly trained MLP networks is required for this classification (Nθ0 networks in Figure 3b).

Therefore, the implementation of A2 is accompanied by an additional classification for the studied buildings. This additional classification can possibly add errors to the entire procedure. On the other hand, A1 is free of these errors and is more flexible, because it can be implemented regardless of the existence of classification of buildings for θ = 0.

Finally, the conclusion regarding the greater efficiency of A2 is also confirmed by means of the CMs (i.e., the corresponding values of the R-index and P-index), which are summarized in Figure 14.

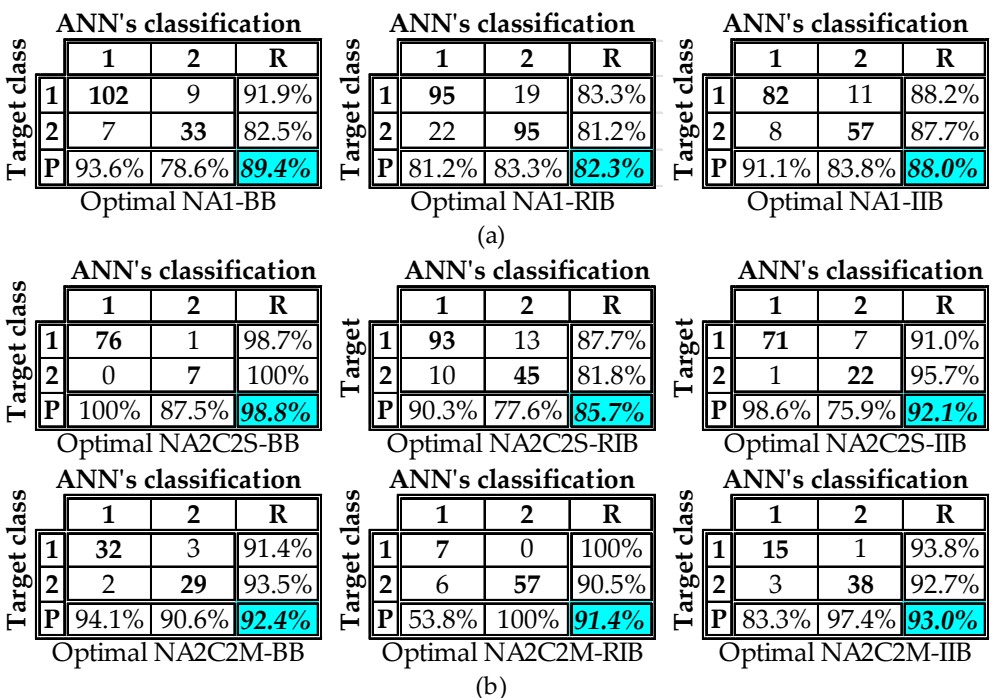

**Figure 14.** CMs extracted by the optimally configured networks of (**a**) A1 and (**b**) A2.

## 5. Conclusions

In this paper, we studied the ability of multilayered feedforward perceptron (MLP) networks to reliably predict the influence of the critical angle of seismic excitation $\theta_{cr}$ on the seismic damage to RC buildings in real time. To this end, the problem was defined and solved as a pattern recognition (PR) problem. Thus, three different seismic damage classes (SDCs) were defined ("slight" or "S", "moderate" or "M", and "heavy" or "H") on the basis of the maximum interstory drift ratio (MIDR), which is a widely used seismic damage index. Two different approaches for the definition of the PR problem were considered. The first (Approach 1, or A1) is a two-class PR problem where the first class corresponds to buildings for which the SDC for $\theta = 0$ is not increased for $\theta = \theta_{cr}$ {Class 1: SDC ($\theta = \theta_{cr}$) = SDC ($\theta = 0$)}, whereas the second class corresponds to the opposite condition {Class 2: SDC ($\theta = \theta_{cr}$) > SDC ($\theta = 0$)}. The second approach (Approach 2, or A2) is based on the initial separation of buildings into two categories, namely, the buildings that are classified to the "S" SDC for $\theta = 0$, and those that are classified to the "M" SDC for $\theta = 0$. Then, two different categories of A2 are defined: in the first framework (A2/C3S), three classes are considered for the buildings that are classified to the "S" SDC for $\theta = 0$ {Class 1: SDC ($\theta = 0$) = "S" → SDC ($\theta = \theta_{cr}$) = "S", Class 2: SDC ($\theta = 0$) = "S" → SDC ($\theta = \theta_{cr}$) = "M", Class 3: SDC ($\theta = 0$) = "S" → SDC ($\theta = \theta_{cr}$) = "H"}, whereas in the second framework (A2/C2S) two classes for these buildings are defined {Class 1: SDC ($\theta = 0$) = "S" → SDC ($\theta = \theta_{cr}$) = "S", Class 2: SDC ($\theta = 0$) = "S" → SDC ($\theta = \theta_{cr}$) = "M" or "H"}. On the other hand, two classes for the buildings that are classified to the "M" SDC for $\theta = 0$ (category A2/C2M) are considered in all cases {Class 1: SDC ($\theta = 0$) = "M" → SDC ($\theta = \theta_{cr}$) = "M", Class 2: SDC ($\theta = 0$) = "M" → SDC ($\theta = \theta_{cr}$) = "H"}. Thus, two types of analysis are defined in the framework of A2—"A2/(C3S + C2M)" and "A2/(C2S + C2M)"—whereas one type of analysis was defined in the framework of A1 (the analysis type "A1").

For each of the above-mentioned types of analysis, one category (four in total) of MLP networks (NA1 networks for "A1", NA2C3S networks for "A2/C3S", NA2C2S networks for "A2/C2S", and NA2C2M networks for "A2/C2M") was configured and trained using training datasets that were generated by the authors in previous published research on the basis of 3 different versions of 30 properly selected RC buildings (Bare Buildings "BB", Regularly Infilled Buildings "RIB", and Irregularly Infilled Buildings "IIB"). As regards the

input vectors for the MLP networks, 12 widely used seismic parameters and 3 structural parameters were used. Due to the fact that the scope of the present research is the study of the influence of the critical angle of seismic excitation on the seismic damage, we used the min/max ratios of the values of the selected seismic parameters that correspond to the two recorded horizontal components of excitations, as opposed to of the values extracted using a numerical combination of them. As regards the three structural parameters, the selection of them was also based on the nature of the problem, as well as on general (and macroscopic) indices that influence the seismic damage response of buildings. Extended parametric investigation was performed for the detection of the optimal configuration of each of the four categories of the MLP networks. In this investigation, the number of hidden layers, the number of neurons in each hidden layer, the types of activation functions of neurons, and the training algorithms were examined. Thus, this parametric investigation led to the optimally configured network of each of the four categories. The performance of the studied configurations was evaluated using the overall accuracy (OA) index. In cases in which this index failed to lead to clear conclusions, the other metrics defined in the framework of the confusion matrices (CMs)—i.e., the R-index and the P-index—were also used. The evaluation of networks was also based on the testing subsets that formed part of the datasets used for the investigation of their generalization ability.

The main conclusions extracted from the above-described research procedure are the following:

- Between the two different forms of A2 for the buildings classified to the SDC "S" for $\theta = 0$, "A2/C2S" was proven more efficient than "A2/C3S" for all versions of the studied RC buildings (i.e., BB, RIB, and IIB). This conclusion was based on comparisons of the values of the OA index, as well as on comparisons of the values of the R- and P-indices. The superiority of "A2/C2S" against "A2/C3S" means that the trained networks in the present study are more efficient for correct classifications in PR problems with two categories. This conclusion (which certainly cannot be characterized as being generally valid) must be further examined in future extended research. However, the low efficiency of the trained networks used in the present study to classify the testing samples to correct SDC eliminates the possibility of reliable predictions about the specific SDC of buildings for $\theta = \theta_{cr}$. Therefore, the reliable predictions concern the information about the change (or lack thereof) in the SDC for $\theta = \theta_{cr}$. Thus, the category "A2/C2S" in combination with the category "A2/C2M" (which corresponds to buildings that are classified to the SDC "M" for $\theta = 0$)—i.e., the analysis type "A2/(C2S + C2M)"—was used for the comparison of A2 with A1;

- A2, expressed in the form of analysis type "A2/(C2S + C2M)", was proven to be more efficient than A1. However, the percentages of the correct classifications extracted by A1 cannot be characterized as unacceptable, since the corresponding OA values were in all cases greater than 80%. On the other hand, the OA values extracted by "A2/(C2S + C2M)" were greater than 90%. The real difference between the two approaches can be reduced, since the application of A2 requires the knowledge of the SDC of buildings for $\theta = 0$. This knowledge can be obtained either by implementation of NTHA, or by the simulation of networks properly trained to predict the SDC of buildings for $\theta = 0$. In both cases, the possibility of the insertion of errors can lead to incorrect data for the implementation of "A2/(C2S + C2M)". On the other hand, A1 can be implemented without the knowledge of the SDC of buildings for $\theta = 0$; thus, A1 is not affected by these additional errors. For this reason, the two approaches can be generally characterized as almost equal;

- As regards the optimal configuration of networks, it was observed that the addition of a second hidden layer improved their classification ability in all studied cases. However, the increases in the OA index values achieved using two hidden layers instead of one are not always significant. On the other hand, the addition of the second hidden layer significantly increases the values of the R- and P-indices in all cases. The optimal number of neurons in the hidden layers cannot be estimated

without the implementation of parametric investigation using a predefined rule; this conclusion is consistent with the findings of the available relative literature. The resilient backpropagation (RP) algorithm was proven to be more effective in the training of NA1 networks (A1). On the other hand, no clear conclusion can be extracted for the analysis of A2, because the RP algorithm and the scaled conjugate gradient (SCG) algorithm (which was also used in the present study) were proven to be more effective in almost the same number of cases belonging to this approach. Finally, it was proven that the introduction of the hyperbolic tangent (tansig) function as the activation function of neurons in the output layer of networks leads to optimal classifications in all analyses using t A1, and in the vast majority of analyses using A2.

The benefit of the present investigation for civil engineers is the ability to extract a rapid prediction as regards the sensitivity of a selected structural system for new RC buildings to the angle of seismic excitation at the stage of design. The same is also valid in the investigation of several strengthening schemes for existing RC buildings prone to extended seismic damage. In other words, the proposed method gives a calculation tool that contributes to the optimization of the design of new RC buildings' structural systems, or to the optimization of the configuration of strengthening schemes for existing RC buildings.

Finally, it must be noted that a future expansion of the present investigation could include the examination of different seismic or/and structural parameters for the input vectors of networks, sensitivity analyses for the estimation of the parameters that most influence the proposed method's efficiency, and the prediction of the value of the critical angle of the seismic excitation using the MLP networks (or other types of network) as calculation tools for regression analysis.

**Author Contributions:** K.M. designed the selected buildings and performed the training of the Artificial Neural Networks. K.K. processed the selected earthquakes and performed the NTHA analyses for the calculation of the Seismic Damage Indices. All authors have read and agreed to the published version of the manuscript.

**Funding:** This research received no external funding.

**Institutional Review Board Statement:** Not applicable.

**Informed Consent Statement:** Not applicable.

**Data Availability Statement:** The raw data supporting the conclusions of this article may be made available upon request from the correspondence author.

**Conflicts of Interest:** The authors declare no conflict of interest.

## Appendix A. Generation of Training Dataset: Structural Parameters

In Appendix A, a brief description of the data that were used for the calculation of the structural parameters of the ANNs' input vectors (parameters StIP1, StIP2, and StIP3) and output vectors (MIDR values) is presented.

In Figures A1 and A2, the configurations in the plans of the selected RC buildings are illustrated. The modeling assumptions that were taken into consideration are compatible with the provisions of EN1998-1 [16] and EN1992-1-1 [56]. More specifically, the assumptions considered for the linear analysis and design of buildings were the following:

- The buildings were considered to be fully fixed to the ground;
- The infill walls were considered only as vertical loads and not as seismic-resistant structural elements;
- The buildings were designed as medium ductility class (MDC) structures [16];
- The behavior factor q was determined according to the recommendations of EN1998-1 [16];
- The buildings were analyzed using the modal response spectrum method;

- For the design of RC members, the load combinations 1.35G + 1.50Q and G + 0.3Q ± E were taken into consideration (G is the dead load, Q is the live load, and E is the seismic load expressed by the simultaneous application of the design spectrum of EN1998-1 [16] for seismic zone II and site class C along the x- and y-axes).

As regards the modeling of the selected RC buildings for the NTHA (calculation of their damage index, i.e., MIDR value), the following assumptions were considered:

- The nonlinear behavior of the RC members was modeled by means of a lumped plasticity model at the column and beam ends, as well as at the bases of the RC walls;
- The material inelasticity of the RC members was modeled using the modified Takeda hysteresis rule (see e.g., [67]) (Figure A3);
- The effects of the axial load–biaxial bending moments interactions at the column and wall hinges were taken into account by using the N-M2-M3 interaction diagram, which was implemented in the software adopted for the application of the analyses [55] (Figure A3);
- As regards the nonlinear modeling of the infill walls in case of RIB and IIB, nonlinear diagonal struts based on the model proposed by Crisafulli [68] were adopted.

Regarding the evaluation of the used damage index MIDR (which corresponds to the maximum drift among the perimeter frames), the post-processing of the NTHA results leading to the calculation of its values is summarized for the case of an n-story building with an arbitrary plan view in Figure A4.

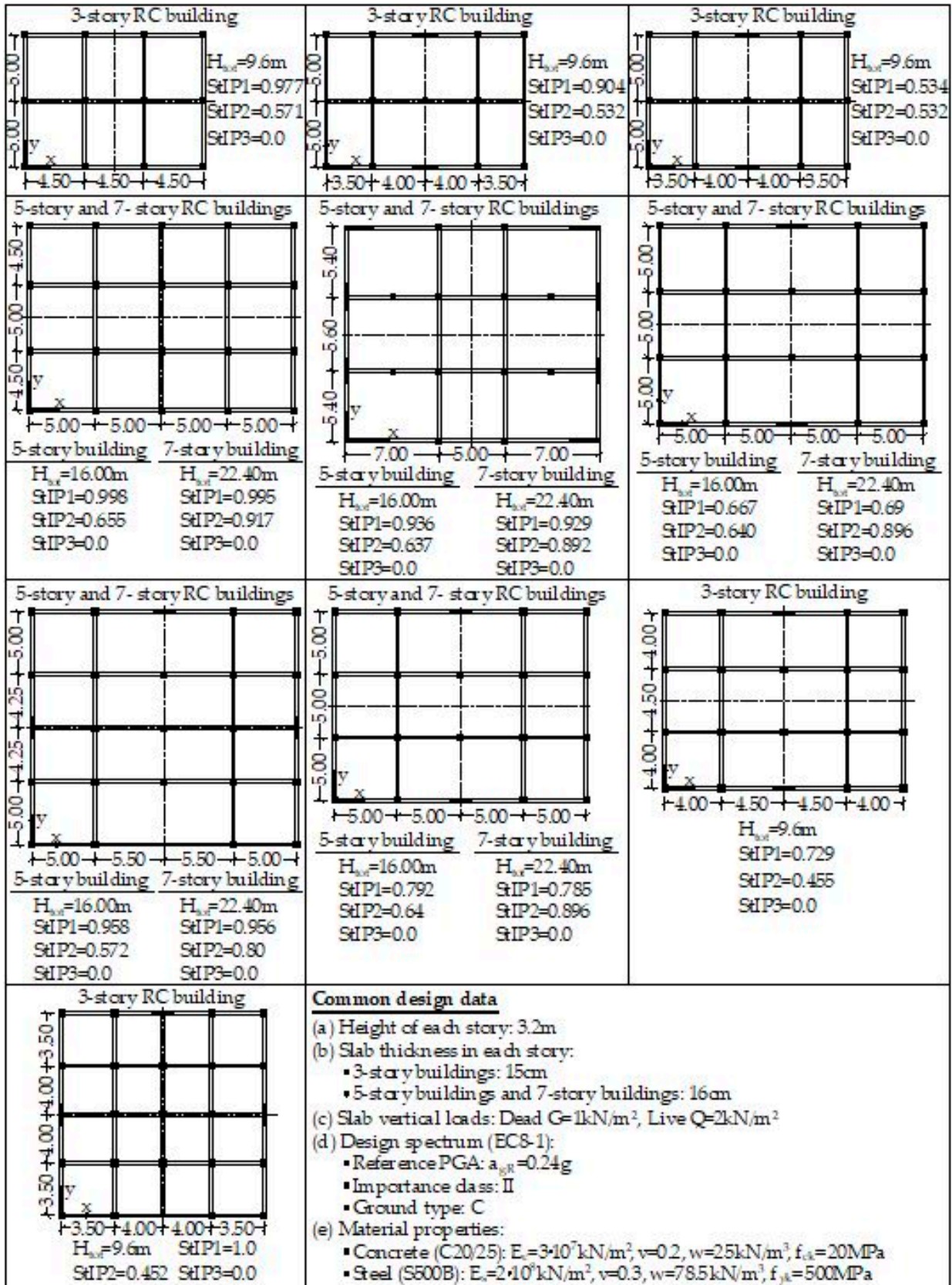

**Figure A1.** Design data of the 15 selected symmetric RC buildings.

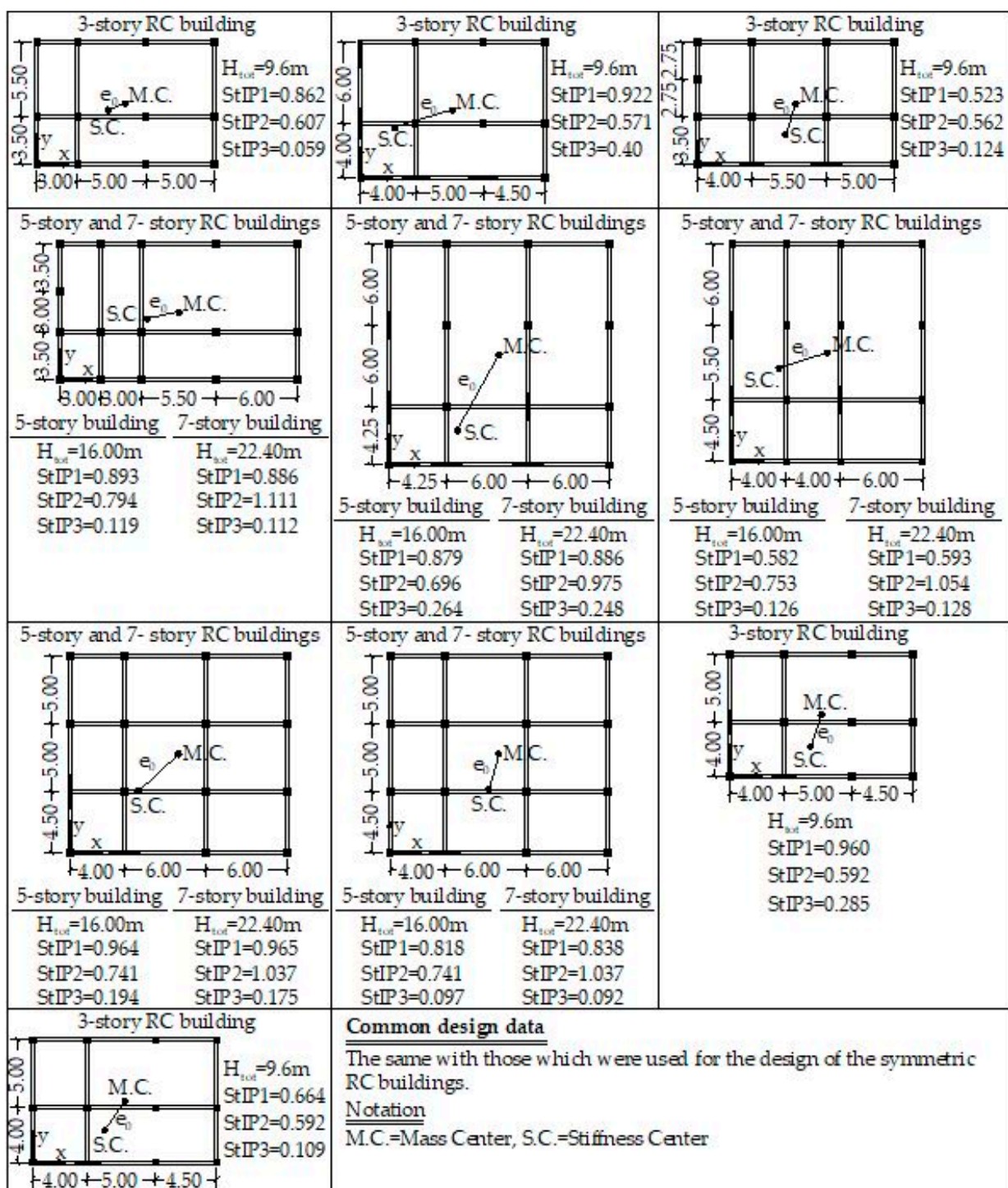

**Figure A2.** Design data of the 15 selected asymmetric RC buildings.

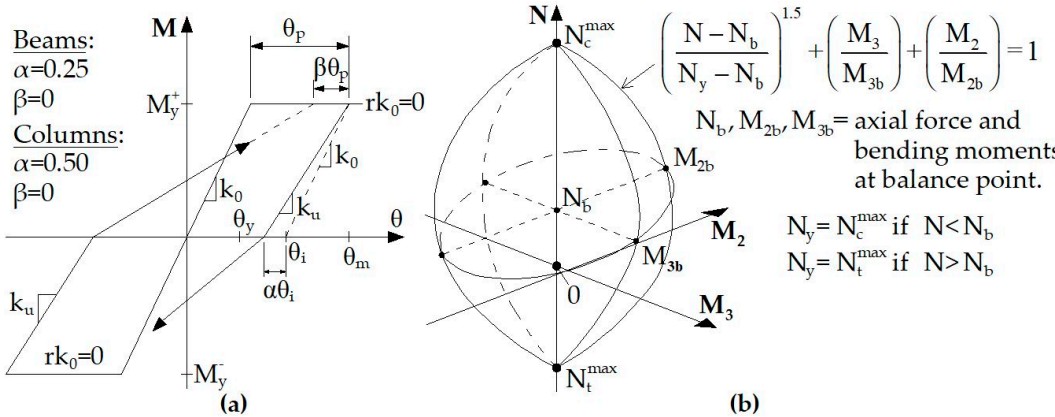

**Figure A3.** (**a**) Moment (M)–Rotation (θ) relationship and (**b**) N–$M_2$–$M_3$ interaction diagram.

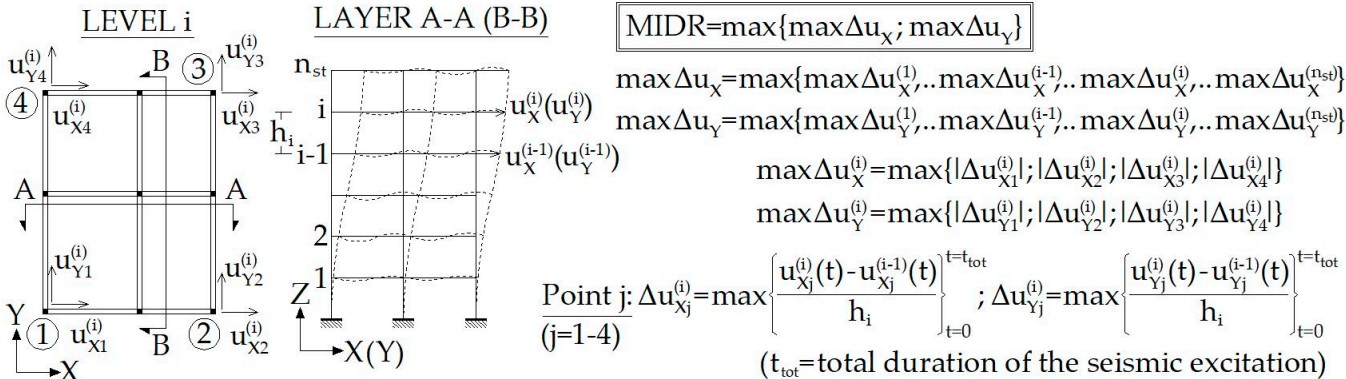

**Figure A4.** Calculation of the MIDR value in the case of an n-story building with arbitrary plan view.

## Appendix B. Generation of Training Dataset: Seismic Parameters

In Appendix B (Table A1), the data of 65 earthquakes that were used for the generation of the ANNs' training dataset are presented.

**Table A1.** Data of the 65 selected seismic excitations.

| No | Earthquake Name | Date | Magnitude (Ms) | Distance to Fault (km) | Component (deg) | PGA (g) |
|----|-----------------|------|----------------|------------------------|-----------------|---------|
| 1 | Imperial Valley | 15 October 1979 | 6.9 | 23.8 | 225/315 | 0.128/0.078 |
| 2 | Imperial Valley | 15 October 1979 | 6.9 | 28.7 | 012/282 | 0.27/0.254 |
| 3 | Kocaeli, (Turkey) | 17 August 1999 | 7.8 | 144.6 | 090/180 | 0.06/0.049 |
| 4 | Landers | 28 June 1992 | 7.4 | 128.3 | 000/270 | 0.057/0.046 |
| 5 | Loma Prieta | 18 October 1989 | 7.1 | 28.2 | 090/180 | 0.247/0.215 |
| 6 | Whittier Narrows | 1 October 1987 | 5.7 | 25.2 | 000/090 | 0.221/0.124 |
| 7 | Northridge | 17 January 1994 | 6.7 | 25.4 | 177/267 | 0.357/0.206 |
| 8 | Northridge | 17 January 1994 | 6.7 | 30 | 020/110 | 0.474/0.439 |
| 9 | N. Palm Springs | 8 July 1986 | 6 | 43.3 | 270/360 | 0.144/0.132 |
| 10 | Northridge | 17 January 1994 | 6.7 | 13 | 000/270 | 0.41/0.482 |
| 11 | Northridge | 17 January 1994 | 6.7 | 6.4 | 090/360 | 0.604/0.843 |
| 12 | Northridge | 17 January 1994 | 6.7 | 12.3 | 000/090 | 0.303/0.443 |
| 13 | Whittier Narrows | 1 October 1987 | 5.7 | 10.8 | 048/318 | 0.426/0.443 |
| 14 | Cape Mendocino | 25 April 1992 | 7.1 | 9.5 | 000/090 | 0.59/0.662 |
| 15 | Chi-Chi (Taiwan) | 20 September 1999 | 7.6 | 2.94 | N/W | 0.251/0.202 |

**Table A1.** *Cont.*

| No | Earthquake Name | Date | Magnitude (Ms) | Distance to Fault (km) | Component (deg) | PGA (g) |
|----|----------------|------|----------------|------------------------|-----------------|---------|
| 16 | Chi-Chi (Taiwan) | 20 September 1999 | 7.6 | 10.04 | N/W | 0.393/0.742 |
| 17 | Chi-Chi (Taiwan) | 20 September 1999 | 7.6 | 4.01 | N/W | 0.162/0.134 |
| 18 | Chi-Chi (Taiwan) | 20 September 1999 | 7.6 | 7.31 | N/W | 0.821/0.653 |
| 19 | Chi-Chi (Taiwan) | 20 September 1999 | 7.6 | 11.14 | N/W | 0.44/0.353 |
| 20 | Chi-Chi (Taiwan) | 20 September 1999 | 7.6 | 10.33 | N/W | 0.13/0.147 |
| 21 | Chi-Chi (Taiwan) | 20 September 1999 | 7.6 | 5.92 | N/W | 0.188/0.148 |
| 22 | Erzincan (Turkey) | 13 March 1992 | | 2.0 | NS/EW | 0.515/0.496 |
| 23 | Loma Prieta | 18 October 1989 | 7.1 | 12.7 | 000/090 | 0.367/0.322 |
| 24 | Loma Prieta | 18 October 1989 | 7.1 | 14.4 | 000/090 | 0.555/0.367 |
| 25 | Loma Prieta | 18 October 1989 | 7.1 | 14.5 | 000/090 | 0.529/0.443 |
| 26 | Northridge | 17 January 1994 | 6.7 | 7.1 | 090/360 | 0.583/0.59 |
| 27 | Northridge | 17 January 1994 | 6.7 | 8.9 | 270/360 | 0.753/0.939 |
| 28 | Northridge | 17 January 1994 | 6.7 | 14.6 | 000/090 | 0.877/0.64 |
| 29 | Northridge | 17 January 1994 | 6.7 | 6.2 | 052/142 | 0.612/0.897 |
| 30 | Campano Lucano (Italy) | 23 November 1380 | 6.9 | 39 | E-W/N-S | 0.047/0.048 |
| 31 | Spitak (Armenia) | 7 December 1988 | 6.7 | 20 | E-W/N-S | 0.183/0.183 |
| 32 | Izmit (Turkey) | 17 August 1999 | 7.6 | 29 | W-E/S-N | 0.129/0.091 |
| 33 | Duzce (Turkey) | 12 November 1999 | 7.2 | 18 | E-W/N-S | 0.8/0.745 |
| 34 | Duzce (Turkey) | 12 November 1999 | 7.2 | 113 | S-N/E-W | 0.022/0.021 |
| 35 | Duzce (Turkey) | 12 November 1999 | 7.2 | 98 | 030/120 | 0.018/0.016 |
| 36 | Duzce (Turkey) | 12 November 1999 | 7.2 | 94 | E-W/N-S | 0.042/0.041 |
| 37 | Izmit (Turkey) | 17 August 1999 | 7.6 | 80 | E-W/N-S | 0.114/0.11 |
| 38 | Duzce (Turkey) | 6 June 2000 | 6.1 | 158 | LONG/TRAN | 0.004/0.004 |
| 39 | Strofades (Greece) | 18 November 1997 | 6.6 | 54 | 261/351 | 0.053/0.054 |
| 40 | Aigion (Greece) | 15 June 1995 | 6.5 | 138 | 065/155 | 0.013/0.013 |
| 41 | Friuli (Italy) | 11 September 1976 | 5.5 | 7 | E-W/N-S | 0.105/0.23 |
| 42 | Volvi (Greece) | 4 July 1978 | | 15 | E-W/N-S | 0.099/0.115 |
| 43 | Dinar (Turkey) | 1 October 1995 | 6.4 | 0 | W-E/S-N | 0.319/0.273 |
| 44 | Izmit (Turkey) | 17 August 1999 | 7.6 | 5 | E-W/N-S | 0.244/0.296 |
| 45 | Duzce (Turkey) | 12 November 1999 | 7.2 | 0 | W-E/S-N | 0.513/0.377 |
| 46 | Imperial Valley | 15 October 1979 | 6.9 | 43.6 | 262/352 | 0.238/0.351 |
| 47 | Loma Prieta | 18 October 1989 | 7.1 | 16.1 | 000/090 | 0.417/0.212 |
| 48 | Loma Prieta | 18 October 1989 | 7.1 | 77.4 | 180/270 | 0.195/0.244 |
| 49 | Northridge | 17 January 1994 | 6.7 | 30.9 | 155/245 | 0.465/0.322 |
| 50 | Northridge | 17 January 1994 | 6.7 | 36.9 | 090/180 | 0.29/0.264 |
| 51 | Duzce, Turkey | 12 November 1999 | 7.3 | 17.6 | 000/090 | 0.728/0.822. |
| 52 | Northridge | 17 January 1994 | 6.7 | 32.7 | 090/180 | 0.103/0.186 |
| 53 | Imperial Valley | 15 October 1979 | 6.9 | 54.1 | 075/345 | 0.122/0.167 |
| 54 | Superstition Hills | 24 November 1987 | 6.6 | 18.2 | 225/315 | 0.156/0.116 |
| 55 | Duzce (Turkey) | 12 November 1999 | 7.3 | 8.2 | 180/270 | 0.348/0.535 |
| 56 | Imperial Valley | 15 October 1979 | 6.9 | 7.6 | 002/092 | 0.213/0.235 |
| 57 | Imperial Valley | 15 October 1979 | 6.9 | 4.2 | 140/230 | 0.485/0.36 |
| 58 | Imperial Valley | 15 October 1979 | 6.9 | 1 | 140/230 | 0.519/0.379 |
| 59 | Imperial Valley | 15 October 1979 | 6.9 | 1 | 140/230 | 0.41/0.439 |
| 60 | Livermore | 27 January 1980 | 5.5 | 3.6 | 270/360 | 0.258/0.233 |
| 61 | Superstition Hills | 24 November 1987 | 6.6 | 13.9 | 000/090 | 0.358/0.258 |
| 62 | Superstition Hills | 24 November 1987 | 6.6 | 13.3 | 090/180 | 0.172/0.211 |
| 63 | Morgan Hill | 24 April 1984 | 6.1 | 12.8 | 270/360 | 0.224./0.348 |
| 64 | Imperial Valley | 15 October 1979 | 6.9 | 12.6 | 140/230 | 0.364/0.38 |
| 65 | Morgan Hill | 24 April 1984 | 6.1 | 3.4 | 150/240 | 0.156/0.312 |

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
