# Peer review of "Rapid Prediction of Seismic Incident Angle’s Influence on the Damage Level of RC Buildings Using Artificial Neural Networks"

_applsci, doi:10.3390/app12031055_

Round 1
Reviewer 1 Report
The manuscript proposes a strategy based on neural networks predicting the influence of the seismic critical input angle on the response of reinforced concrete buildings. The strategy uses a damage classification performed by means of maximum inter-story drift. Moreover, twelve seismic parameters and three structural parameters, are employed to infer the structural response of each considered building.
The manuscript is consistent with the aims and scopes of Applied Sciences. In general, it is well written although the use of acronyms is quite exaggerated so that comprehension of the manuscript may be difficult. Moreover, the reviewer has a few of significant concerns that should be addressed before that it can be accepted for publication.
As a first comment, the manuscript presents a specific section discussing the theoretical background as well as the historical evolution of the seismic codes and of the common practice. Actually, if such aspects, not directly related to the presented strategy, are discussed, the state of the art is significantly deficient. In particular, it is essential to make reference to the CQC3 modal combination rule developed in Smeby and Der Kiureghian (1985) and Menun and Der Kiureghian (1998) because of its importance, as well as its applications in investigating the effects of the critical angle (see Lopez et al. 2000, 2001) and its applications concerning reinforced concrete structures (see Sessa et al. 2015, 2021).
Lines 241-245. Sub-spaces are subsets of a parent space inheriting its topology and its characteristic operations. In such a case, the value SDIMFc seems to be a sort of threshold splitting a numerical domain into two sub-domains. Hence, the use of sub-space is incorrect. Please, check or clarify.
The main concern is about the aims of the proposed procedure and its scientific advancements. Actually, the authors claim that such a procedure may be used for expeditive predictions about the influence of the critical angle in case of design of new structures or seismic assessment of existing ones. Actually, the design of new structures can be performed by accounting the effects of the critical angle, as specified in the previous comment, by using the CQC3 combination rule and its derived algorithms such as seismic envelopes. Actually, the use of a neural network for determining if and how the designer should account for the critical angle, seems to be a trivial issue. Actually, the case of existing buildings analyzed by time history analyses may find a few benefits by the procedure, but still the use of a neural network, which must be trained, for avoiding to perform a few analyses seems a kind of overshooting. In this sense, the procedure seems to be quite speculative and the manuscript does not highlights any significant scientific advancement with respect to the state of the art.
A further issue concerns the fundamentals of the procedure in Equation 1. Actually, the whole classification is performed by comparing the structural performance associated with an earthquake acting along the “original” structural axes (SDI(theta=0)) with those relevant to an excitation acting along the critical angle (SDI(theta = theta_cr)). As a matter of fact, the “original” structural axes are completely arbitrary. Hence, the classification of a structural model could be significantly different if different axes are adopted. As an example, if one chooses the critical direction as structural axes, the SDIMF would be always unitary this affecting the classification of the building. Authors should discuss such an issue and provide evidence that it does not affects the whole procedure.
A similar issue arises by Equation 3 in which the “uncoupled fundamental natural periods […] along their structural axes X, Y” are used to define the StIP1 parameter. Again, axes are arbitrary but, most important, in common practice structures vibration modes are not necessarily uncoupled and may be oriented along different directions. Again, authors should discuss such an issue and provide evidence that it does not affects the whole procedure.
As a minor comment, the use of acronyms as variables is very confusing. E.g., it is very hard to understand the approaches described in line 262-298. Please, consider to adopt a more concise nomenclature.
REFERENCES
Lopez, O.A., Chopra, A.K., Hernandez, J.J. Critical response of structures to multicomponent earthquake excitation (2000) Earthquake Engineering and Structural Dynamics, 29 (12), pp. 1759-1778.
López, O.A., Chopra, A.K., Hernández, J.J. Evaluation of combination rules for maximum response calculation in multicomponent seismic analysis (2001) Earthquake Engineering and Structural Dynamics, 30 (9), pp. 1379-1398.
Menun, C., Der Kiureghian, A. A replacement for the 30%, 40%, and SRSS rules for multicomponent seismic analysis (1998) Earthquake Spectra, 14 (1), pp. 153-163.
Sessa, S., Marmo, F., Rosati, L. Effective use of seismic response envelopes for reinforced concrete structures (2015) Earthquake Engineering and Structural Dynamics, 44 (14), pp. 2401-2423.
Sessa, S., Marmo, F., Vaiana, N., Rosati, L. A Computational Strategy for Eurocode 8-Compliant Analyses of Reinforced Concrete Structures by Seismic Envelopes (2021) Journal of Earthquake Engineering, 25 (6), pp. 1078-1111.
Smeby, W., der Kiureghian, A. Modal combination rules for multicomponent earthquake excitation (1985) Earthquake Engineering & Structural Dynamics, 13 (1), pp. 1-12.
Author Response
Our replies to your comments are included in the attached pdf file which is also includes the revised paper.

Reviewer 2 Report
The manuscript is of high scientific quality and proves extensive research and work of the authors. Nevertheless, it is my opinion that the engineering significance must be better emphasized. There is an abundance of details on the algorithms used, and few details on the structural earthquake engineering methods and metrics used. It is my belief that the general readership of the paper will consist in civil engineers, thus details on selecting ground motions, input parameters and models employed in NTHAs, processing of output data shall be at stake. Moreover, the manuscript shall highlight that the ground motions are as recorded and are rotated with respect to the axis of the building. The problem around is to rotate the as recorded ground motion components and apply the rotated components along the principal axes of the building. Other approach would be to apply the principal (maximum and minimum) rotated components to the building oriented at different incident angles with respect to these components. These are my general comments. Some minor comments are included in the uploaded file.

Author Response

(The authors gave the same response as above.)

Round 2
Reviewer 1 Report
The Authors addressed all the major issues reported in the first reviews. Hence, the manuscript can be accepted for publication in the present form.